# Deep Regression Representation Learning with Topology

## Abstract

The information bottleneck (IB) principle is an important framework that provides guiding principles for representation learning. Most works on representation learning and the IB principle focus only on classification and neglect regression. Yet the two operate on different principles to align with the IB principle: classification targets class separation in feature space, while regression requires feature continuity and ordinality with respect to the target. This key difference results in topologically different feature spaces. Why does the IB principle impact the topology of feature space? In this work, we establish two connections between them for regression representation learning. The first connection reveals that a lower intrinsic dimension of the feature space implies a reduced complexity of the representation $\mathbf{Z}$, which serves as a learning target of the IB principle. This complexity can be quantified as the entropy of $\mathbf{Z}$ conditional on the target space $\mathbf{Y}$, and it is shown to be an upper bound on the generalization error. The second connection suggests that to better align with the IB principle, it's beneficial to learn a feature space that is topologically similar to the target space. Motivated by the two connections, we introduce a regularizer named PH-Reg, to lower the intrinsic dimension of feature space and keep the topology of the target space for regression. Experiments on synthetic and real-world regression tasks demonstrate the benefits of PH-Reg.

## 1 Introduction

Regression is a fundamental task in machine learning in which input samples are mapped to a continuous target space. Representation learning is important for regression as it empowers models to automatically extract, transform, and leverage relevant information from data, leading to improved performance. The information bottleneck principle (Shwartz-Ziv & Tishby, 2017) provides a theoretical framework and guiding principle for learning the representation. It suggests that neural network aims to learn a representation $\mathbf{Z}$ which contains sufficient information about the target $\mathbf{Y}$ but minimal information about the input $\mathbf{X}$. For representation $\mathbf{Z}$, the sufficiency retains the necessary information about $\mathbf{Y}$, while the minimality reduces $\mathbf{Z}$'s complexity and prevents overfitting. The optimal representation, as specified by Achille & Soatto (2018a;b), is the most useful(sufficient), minimal, and invariant to nuisance factors, and the minimality is deeply linked to the invariance. However, the studies of (Achille & Soatto, 2018a;b) are only specified for classification. In fact, many works study representation learning from a classification point of view (Ma et al., 2018; Zhu et al., 2018) but ignore the equally important task of regression.

While both regression and classification follow the minimal and sufficient representation learning target as suggested by the IB principle, there are some fundamental differences. For example, regression representations are commonly continuous and form an ordinal relationship to align with the IB principle (Zhang et al., 2023). By contrast, classification shortens the distance of features belonging to the same class to learn minimal representation and increases the distance of features belonging to different classes to learn sufficient representation (Boudiaf et al., 2020), which leads to disconnected representations (Brown et al., 2022a). The continuity represents the $0^{th}$ Betti number in topology, influencing the 'shape' of the feature space. We thus wonder what the connections are between the topology of the feature space and the IB principle for regression representation learning.

In this work, we establish two connections between the topology of the feature space and the IB principle for regression representation learning. To establish the connections, we first demonstrate

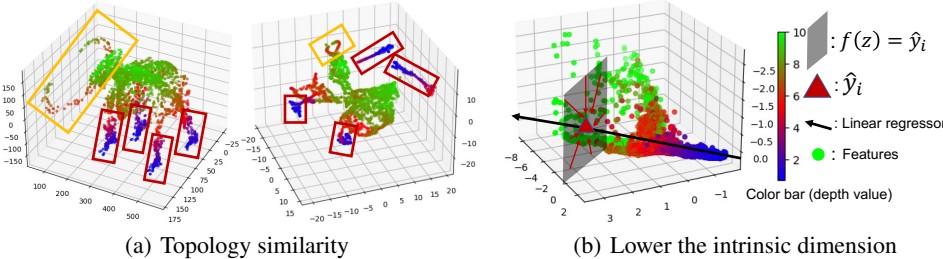

(a) Topology similarity

(b) Lower the intrinsic dimension

Figure 1: (a) Visualization of the feature space (left) and the 'Mammoth' shape target space (right), see Sec. 5.1 for details. The feature space is topologically similar to the target space . (b) Visualization of the feature space from depth estimation task. As shown in the gray quadrilateral, enforcing a lower intrinsic dimension can reduce the unnecessary $\mathcal{H}(\mathbf{Z}|\mathbf{Y} = \mathbf{y}_i)$ corresponding to the solution space of $f(\mathbf{z}) = \hat{\mathbf{y}}_i$. Here, $\hat{\mathbf{y}}_i$ is the predicted depth and the black arrow is a linear regressor.

that the IB principle can be formulated as an optimization problem minimizing both $\mathcal{H}(\mathbf{Y}|\mathbf{Z})$ and $\mathcal{H}(\mathbf{Z}|\mathbf{Y})$. Specifically, for regression, the $\mathcal{H}(\mathbf{Z}|\mathbf{Y})$ is linked to the minimality of $\mathbf{Z}$ and serves as an upper-bound on the generalization error.

The first connection suggests that decreasing the intrinsic dimension of the feature space results in a lower $\mathcal{H}(\mathbf{Z}|\mathbf{Y})$, indicating improved generalization performance. The intrinsic dimension is a fundamental topology property of data representation, which can be regarded as the minimal number of coordinates to describe the representation without significant information loss (Ansuini et al., 2019; Gong et al., 2019). Figure 1(b) provides a visualization of the feature space for depth estimation. In this figure, the predicted depth $\hat{\mathbf{y}}$ is obtained by mapping the features (represented as dots) to the black arrow (indicating a linear regressor). The gray quadrilateral in Figure 1(b) represents the solution space of $f(\mathbf{z}) = \hat{\mathbf{y}}_i$, which is closely related to the $\mathcal{H}(\mathbf{Z}|\mathbf{Y} = \mathbf{y}_i)$. Enforcing a lower intrinsic dimension can encourage this solution space squeezed to be a point, which implies a lower $\mathcal{H}(\mathbf{Z}|\mathbf{Y} = \mathbf{y}_i)$. Encourage a lower $\mathcal{H}(\mathbf{Z}|\mathbf{Y} = \mathbf{y}_i)$ for all the $i$ will result in a lower $\mathcal{H}(\mathbf{Z}|\mathbf{Y})$. The first connection suggests learning a lower intrinsic dimension feature space for a lower $\mathcal{H}(\mathbf{Z}|\mathbf{Y})$.

The second connection shows the representation $\mathbf{Z}$ is homeomorphic to the target space $\mathbf{Y}$ when both the $\mathcal{H}(\mathbf{Y}|\mathbf{Z})$ and the $\mathcal{H}(\mathbf{Z}|\mathbf{Y})$ are minimal. The homeomorphic between two spaces can be intuitively understood as one can be continuously deformed to the other, and in the topology view, two spaces are considered the same if they are homeomorphic. Figure 1(a) provides a t-SNE visualization of the 100-dimensional feature space with a 'Mammoth' shape target space. This feature space is topologically similar to the target space, which indicates regression potentially captures the topology of the target space. The second connection suggests improving such similarity.

These connections naturally inspire us to learn a regression feature space that is topologically similar to the target space while also having a lower intrinsic dimension. To this end, we introduce a regularizer called Persistent Homology Regression Regularizer (PH-Reg). In classification, interest has grown in regulating the intrinsic dimension. For instance, Zhu et al. (2018) explicitly penalizes intrinsic dimension as regularization, while Ma et al. (2018) uses intrinsic dimensions as weights for noise label correction. However, a theoretical justification for using intrinsic dimension as a regularizer is lacking, and they overlook the topology of the target space. Experiments on various regression tasks demonstrate the effectiveness of PH-Reg. Our main contributions are three-fold:

- To our best knowledge, we are the first to explore topology in the context of regression representation learning. We establish novel connections between the topology of the feature space and the IB principle, which also provides justification for exploiting intrinsic dimension as a regularizer.

- Based on the IB principle, we show that reducing $\mathcal{H}(\mathbf{Z}|\mathbf{Y})$ is the key to learning the minimal representation, and it is upper-bound on the generalization error in regression.

- Based on the established connections, we designed a regularizer named PH-Reg, which achieves significant improvement on synthetic datasets for coordinate prediction as well as real-world regression tasks, including super-resolution, age estimation and depth estimation.

## 2 RELATED WORKS

**Intrinsic dimension**. Input data and representations often live in lower intrinsic dimension manifolds but are embedded within a higher-dimensional ambient space (Bengio et al., 2013). The intrinsic dimension of the last hidden layer of a neural network has shown a strong connection with the network generalization ability (Ansuini et al., 2019); commonly, the generalization ability increases with the decrease of the intrinsic dimension. Several widely used regularizers like weight decay and dropout effectively reduce the last hidden layer's intrinsic dimension (Brown et al., 2022b). Among the relevant studies, Birdal et al. (2021) is the most closely related to ours. This work demonstrates that the generalization error can be bounded by the intrinsic dimension of training trajectories, which possess fractal structures, and thus regulating its intrinsic dimension. However, their analysis is based on the parameter space, while ours is on the feature space, and we regulate the intrinsic dimension while preserving the topology of the target space.

**Topological data analysis**. Topological data analysis has recently emerged in machine learning. It can be coupled with feature learning to ensure that learned representations are robust and reflect the training data's underlying topological and geometric information. By doing so, it has benefitted a diverse set of tasks ranging from fMRI data analysis (Rieck et al., 2020) to classification of 3D surface meshes (Reininghaus et al., 2015) and graphs (Zhao & Wang, 2019). Topology-based regularizers aim to control properties like connectivity (Hofer et al., 2019) and topological complexity (Chen et al., 2019). Topology-preserving representations can be learned by preserving 0-dimensional (Moor et al., 2020) and 1-dimensional (Trofimov et al., 2023) topologically relevant distances of the input space and the feature space. We follow these works to preserve topology information. However, unlike classification, regression's target space is naturally a topology space, rich in topology information crucial for the detailed task. Consequently, we leverage the topology of the target space, marking the first exploration of topology in the context of regression representation learning.

## 3 LEARNING DESIRABLE REGRESSION REPRESENTATION

From a topology point of view, what kind of representation is desirable for regression? Or, more simply, what shape or structure should the feature space have for effective regression? In this work, we suggest a desirable regression representation should (1) have a feature space topologically similar to the target space and (2) the intrinsic dimension of the feature space should be the same as the target space. We arrive at this conclusion by establishing connections between the topology of the feature space and the Information Bottleneck principle.

Consider a dataset $S = \{\mathbf{x}_i, \mathbf{y}_i\}_{i=1}^{N}$ with $N$ samples $\mathbf{x}_i$, which typically is an image ($\mathbf{x}_i \in \mathbb{R}^{d_{x_1} \times d_{x_2}}$) or video ($\mathbf{x}_i \in \mathbb{R}^{d_{x_1} \times d_{x_2} \times d_{x_3}}$) in computing vision, sampled from a distribution $P$, and the corresponding label $\mathbf{y}_i \in \mathbb{R}^{d_y}$. To predict $\mathbf{y}_i$, a neural network first encodes the input $\mathbf{x}_i$ to a representation $\mathbf{z}_i \in \mathbb{R}^d$ before apply a regressor $f$, *i.e.*, $\hat{\mathbf{y}}_i = f(\mathbf{z}_i)$. The encoder and the regressor $f$ are trained by minimizing a task-specific regression loss $\mathcal{L}_m$ based on a distance between $\hat{\mathbf{y}}_i$ and $\mathbf{y}_i$, *i.e.*, $\mathcal{L}_m = g(||\hat{\mathbf{y}}_i - \mathbf{y}_i||_2)$. Typically, an L2 loss is used, *i.e.*, $\mathcal{L}_m = \frac{1}{N}\sum_i ||\hat{\mathbf{y}}_i - \mathbf{y}_i||_2$, though more robust variants exist such as L1 or the scale-invariant error (Eigen et al., 2014). Note that the dimensionality of $\mathbf{y}_i$ is task-specific and need not be limited to 1.

We denote $X, Y, Z$ as random variables representing $\mathbf{x}, \mathbf{y}, \mathbf{z}$, respectively. The Information Bottleneck tradeoff is a practical implementation of the IB principle in machine learning. It suggests a desirable representation $\mathbf{Z}$ should contain sufficient information about the target $\mathbf{Y}$, *i.e.*, maximize the mutual information $I(\mathbf{Z}; \mathbf{Y})$, but minimal information about the input $\mathbf{X}$, *i.e.*, minimize $I(\mathbf{Z}; \mathbf{X})$. The tradeoff between the two aims is typically formulated as an optimization problem with the associated Lagrangian to be minimized $\mathcal{IB} := I(\mathbf{Z}; \mathbf{X}) - \beta I(\mathbf{Z}; \mathbf{Y})$, where $\beta > 0$ is the Lagrange multiplier.

To connect the topology of the feature space to the Information Bottleneck principle, we first formulate the IB principle into relationships purely between $\mathbf{Y}$ and $\mathbf{Z}$.

**Theorem 1** *Optimizing the Information Bottleneck trade-off Lagrangian is equivalent to minimizing the conditional entropies $\mathcal{H}(\mathbf{Y}|\mathbf{Z})$ and $\mathcal{H}(\mathbf{Z}|\mathbf{Y})$.*

The detailed proof of Theorem 1 is provided in Appendix A.1. Here, we provide a brief overview of the terms. The conditional entropy $\mathcal{H}(\mathbf{Y}|\mathbf{Z})$ encourages the learned representation $\mathbf{Z}$ to be informative about the target variable $\mathbf{Y}$. When considering $\mathcal{I}(\mathbf{Z}; \mathbf{Y})$ as a signal, the term $\mathcal{H}(\mathbf{Z}|\mathbf{Y})$ in Theorem

1 can be thought of as noise, since it equals the total information $\mathcal{H}(\mathbf{Z})$ minus the signal $\mathcal{I}(\mathbf{Z}; \mathbf{Y})$, consequently, minimizing $\mathcal{H}(\mathbf{Z}|\mathbf{Y})$ can be seen as learning a minimal representation by reducing noise. The minimality can reduce the complexity of $\mathbf{Z}$ and prevent neural networks from overfitting. Below, we show the connection between $\mathcal{H}(\mathbf{Z}|\mathbf{Y})$ and the generalization ability.

**Theorem 2** *We are given dataset $S = \{\mathbf{x}_i, \mathbf{z}_i, \mathbf{y}_i\}_{i=1}^N$ sampled from distribution $P$, where $\mathbf{x}_i$ is the input, $\mathbf{z}_i$ is the corresponding representation, and $\mathbf{y}_i$ is the label. Let $d_{max} = \max_{\mathbf{y} \in \mathcal{Y}} \min_{\mathbf{y}_i \in S} ||\mathbf{y} - \mathbf{y}_i||_2$ be the maximum distance of $\mathbf{y}$ to its nearset $\mathbf{y}_i$. Assume $(\mathbf{Z}|\mathbf{Y} = \mathbf{y}_i)$ follows a disribution $\mathcal{D}$ and the following holds:*

$$\mathbb{E}_{\mathbf{z} \sim \mathcal{D}}[||\mathbf{z} - \bar{\mathbf{z}}||_2] \leq Q(\mathcal{H}(\mathcal{D})), \tag{1}$$

*where $\bar{\mathbf{z}}$ is the mean of the distribution $\mathcal{D}$ and $Q(\mathcal{H}(\mathcal{D}))$ is some function of $\mathcal{H}(\mathcal{D})$. The above implies the dispersion of the distribution $\mathcal{D}$ is bounded by its entropy, which usually is the case, like the multivariate normal distribution and the uniform distribution. Assume the regressor $f$ is $L_1$-Lipschitz continuous, then as $d_{max} \to 0$, we have*

$$\mathbb{E}_{\{\mathbf{x},\mathbf{z},\mathbf{y}\} \sim P}[||f(\mathbf{z}) - \mathbf{y}||_2] \leq \mathbb{E}_{\{\mathbf{x},\mathbf{z},\mathbf{y}\} \sim S}(||f(\mathbf{z}) - \mathbf{y}||_2) + 2L_1 Q(\mathcal{H}(\mathbf{Z}|\mathbf{Y})) \tag{2}$$

**Proposition 1** *If $\mathcal{D}$ is a multivariate normal distribution $\mathcal{N}(\bar{\mathbf{z}}, \Sigma = k\mathbf{I})$, where $k > 0$ is a scalar and $\bar{\mathbf{z}}$ is the mean of the distribution $\mathcal{D}$. Then, the function $Q(\mathcal{H}(\mathcal{D}))$ in Theorem 2 can be selected as $Q(\mathcal{H}(\mathcal{D})) = \sqrt{\frac{d(e^{2\mathcal{H}(\mathcal{D})})^{\frac{1}{d}}}{2\pi e}}$, where $d$ is the dimension of $\mathbf{z}$. If $\mathcal{D}$ is a uniform distribution, then the $Q(\mathcal{H}(\mathcal{D}))$ can be selected as $Q(\mathcal{H}(\mathcal{D})) = \frac{e^{\mathcal{H}(\mathcal{D})}}{\sqrt{12}}$.*

The detailed proof of Theorem 2 and Proposition 1 are provided in Appendix A.2. Theorem 2 states that the generalization error $|\mathbb{E}_P[||f(\mathbf{z}) - \mathbf{y}||_2] - \mathbb{E}_S[||f(\mathbf{z}) - \mathbf{y}||_2]|$, defined as the difference between the population risk $\mathbb{E}_P[||f(\mathbf{z}) - \mathbf{y}||_2]$ and the empirical risk $\mathbb{E}_S[||f(\mathbf{z}) - \mathbf{y}||_2]$, is bounded by the $\mathcal{H}(\mathbf{Z}|\mathbf{Y})$ in Theorem 1. Proposition 1 provides examples of the function $Q$ for various distributions.

Theorem 2 suggests minimizing $\mathcal{H}(\mathbf{Z}|\mathbf{Y})$ will improve generalization performance. Now, we can establish our first connection between the topology of the feature space and the IB principle.

**Theorem 3** *Assume that $\mathbf{z}$ lies in a manifold $\mathcal{M}$ and the $\mathcal{M}_i \subset \mathcal{M}$ is a manifold corresponding to the distribution $(\mathbf{z}|\mathbf{y} = \mathbf{y}_i)$. Assume for all features $\mathbf{z}_i \in \mathcal{M}_i$, the following holds:*

$$\int_{||\mathbf{z}-\mathbf{z}_i|| \leq \epsilon} P(\mathbf{z})d\mathbf{z} = C(\epsilon), \tag{3}$$

*where $C(\epsilon)$ is some function of $\epsilon$. The above imposes a constraint where the distribution $(\mathbf{z}|\mathbf{y} = \mathbf{y}_i)$ is uniformly distributed across $\mathcal{M}_i$. Then, as $\epsilon \to 0^+$, we have:*

$$\mathcal{H}(\mathbf{Z}|\mathbf{Y}) = \mathbb{E}_{\mathbf{y}_i \sim \mathcal{Y}} \mathcal{H}(\mathbf{Z}|\mathbf{Y} = \mathbf{y}_i) = \mathbb{E}_{\mathbf{y}_i \sim \mathcal{Y}}[-\log(\epsilon) Dim_{ID}\mathcal{M}_i + \log \frac{K}{C(\epsilon)}], \tag{4}$$

*for some fixed scalar K. $Dim_{ID}\mathcal{M}_i$ is the intrinsic dimension of the manifold $\mathcal{M}_i$.*

The detailed proof of Theorem 3 is provided in Appendix A.3. Theorem 3 states that if the distribution $(\mathbf{z}|\mathbf{y} = \mathbf{y}_i)$ lies on a manifold $\mathcal{M}_i$ and is uniformly distributed across $\mathcal{M}_i$, then the $\mathcal{H}(\mathbf{Z}|\mathbf{Y})$ is positively related ($-\log(\epsilon) > 0$ as $\epsilon \to 0^+$) to the expected intrinsic dimension of the $\mathcal{M}_i$.

Since $\mathcal{M}_i \subset \mathcal{M}$, Theorem 3 suggests that reducing the intrinsic dimension of the feature space $\mathcal{M}$ will lead to a lower $\mathcal{H}(\mathbf{Z}|\mathbf{Y})$, which in turn implies a better generalization performance based on Theorem 2. On the other hand, the intrinsic dimension of $\mathcal{M}$ should not be less than the intrinsic dimension of the target space to guarantee sufficient representation capabilities. Thus, a $\mathcal{M}$ with an intrinsic dimension equal to the dimensionality of the target space is desirable.

Below, we establish the second connection: topological similarity between the feature and target spaces. We first define the optimal representation following Achille & Soatto (2018b).

**Definition 1** *(Optimal Representation). The representation $\mathbf{Z}$ is optimal if $\mathcal{H}(\mathbf{Y}|\mathbf{Z}) = \mathcal{H}(\mathbf{Z}|\mathbf{Y}) = 0$.*

**Proposition 2** *If the representation $\mathbf{Z}$ is optimal and the mapping $f'$ between $\mathbf{Z}$ and $\mathbf{Y}$ and its inverse $f'^{-1}$ are continuous, then $\mathbf{Z}$ is homeomorphic to $\mathbf{Y}$.*

The detailed proof of the Proposition 2 is provided in Appendix A.4. Proposition 2 shows that the optimal $\mathbf{Z}$ is homeomorphic to $\mathbf{Y}$, which suggests encouraging $\mathbf{Z}$ and $\mathbf{Y}$ to be homeomorphic.

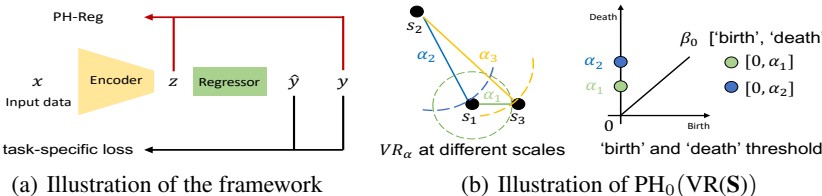

(a) Illustration of the framework $\qquad$ (b) Illustration of $\text{PH}_0(\text{VR}(\mathbf{S}))$

Figure 2: Illustration of the (a) the use of PH-Reg for regression, and (b) calculating of $\text{PH}_0(\text{VR}(\mathbf{S}))$. Here $\mathbf{S} = \{s_1, s_2, s_3\}$. We say three connected components, *i.e.*, $\beta_0$, $(\{\{s_1\}, \{s_2\}, \{s_3\}\})$ 'birth' when $\alpha = 0$, one 'death' (two left $(\{\{s_1, s_3\}, s_2\})$) when $\alpha = \alpha_1$, and another one 'death' (one left $(\{\{s_1, s_3, s_2\}\})$) when $\alpha = \alpha_2$. Thus $\text{PH}_0(\text{VR}(\mathbf{S})) = \{[0, \alpha_1], [0, \alpha_2]\}$.

However, achieving this directly is challenging since the representation $\mathbf{Z}$ typically lies in a high-dimensional space that cannot be modeled without sufficient data samples. As such, we opted to enforce the topological similarity between the target and feature spaces. The two established connections imply that the desired $\mathbf{Z}$ should exhibit topological similarity to the target space and share the same intrinsic dimension as the target space.

# 4 PERSISTENT HOMOLOGY REGRESSION REGULARIZER

Our analysis in Sec. 3 inspires us to learn a lower intrinsic dimension feature space that is topologically similar to the target space. To this end, we propose a regularizer named PH-Reg, which contains an intrinsic dimension term $\mathcal{L}_d$ to lower the intrinsic dimension and a topology term $\mathcal{L}_t$ to encourage the topological similarity. The design of PH-Reg is inspired by the topology autoencoder (Moor et al., 2020) and Birdal's regularizer (Birdal et al., 2021). To better understand the mechanics, we first introduce some preliminaries on topology before outlining our proposed regularizer (Sec. 4.2).

## 4.1 PRELIMINARIES

The simplicial complex is a central object in algebraic topological data analysis, and it can be exploited as a tool to model the 'shape' of data. Given a set of finite samples $\mathbf{S} = \{s_i\}$, the simplicial complex $K$ can be seen as a collection of simplices $\sigma = \{s_0, \cdots, s_k\}$ of varying dimensions: vertices $(|\sigma| = 1)$, edges$(|\sigma| = 2)$, and the higher-dimensional counterparts$(|\sigma| > 2)$. For each $\mathbf{S}$, there exist many ways to build simplicial complexes and the Vietoris-Rips Complexes are widely used:

**Definition 2** *(Vietoris-Rips Complexes). Given a set of finite samples $\mathbf{S}$ sampled from the feature space or target space and a threshold $\alpha \geq 0$, the Vietoris-Rips Complexes $VR_\alpha$ is defined as:*

$$VR_\alpha(\mathbf{S}) = \{\{s_0, \cdots, s_k\}, s \in \mathbf{S} | d(s_i, s_j) \leq \alpha\}, \tag{5}$$

*where $d(s_i, s_j)$ is the Euclidean distance between samples $s_i$ and $s_j$.*

Let $C_k(\text{VR}_\alpha(\mathbf{S}))$ denote the vector space generated by its $k$-dimensional simplices over $\mathbb{Z}_2$ [1]. The boundary operator $\partial_k : C_k(\text{VR}_\alpha(\mathbf{S})) \to C_{k-1}(\text{VR}_\alpha(\mathbf{S}))$, which maps each simplex to its boundary, is a homomorphism between $C_k(\text{VR}_\alpha(\mathbf{S}))$ and $C_{k-1}(\text{VR}_\alpha(\mathbf{S}))$. The $k^{\text{th}}$ homology group $H_k(\text{VR}_\alpha(\mathbf{S}))$ is defined as the quotient group $H_k(\text{VR}_\alpha(\mathbf{S})) := \ker\partial_k/\text{im}\partial_{k+1}$. Rank $H_k(\text{VR}_\alpha(\mathbf{S}))$ is known as the $k^{\text{th}}$ Betti number $\beta_k$, which counts the number of $k$-dimensional holes and can be used to represent the topological features of the manifold that the set of points $\mathbf{S}$ sampled from.

However, the $H_k(\text{VR}_\alpha(\mathbf{S}))$ is obtained based on a single $\alpha$, which is easily affected by small changes in $\mathbf{S}$. Thus it is not robust and is of limited use for real-world datasets. The persistent homology considers all the possible $\alpha$ instead of a single one, which results in a sequence of $\beta_k$. This is achieved through a nested sequence of simplicial complexes, called *filtration*: $\text{VR}_0(\mathbf{S}) \subseteq \text{VR}_{\alpha_1}(\mathbf{S}) \subseteq \cdots \subseteq \text{VR}_{\alpha_m}(\mathbf{S})$ for $0 \leq \alpha_1 \leq \alpha_m$. Let $\gamma_i = [\alpha_i, \alpha_j]$ be the interval corresponding to a $k$-dimensional hole 'birth' at the threshold $\alpha_i$ and 'death' at the threshold $\alpha_j$, we denote $\text{PH}_k(\text{VR}(\mathbf{S})) = \{\gamma_i\}$ the set of 'birth' and 'death' intervals of the $k$-dimensional holes. We only consider $\text{PH}_0(\text{VR}(\mathbf{S}))$ in this work,

---

[1]It is not specific to $\mathbb{Z}_2$, but $\mathbb{Z}_2$ is a typical choice.

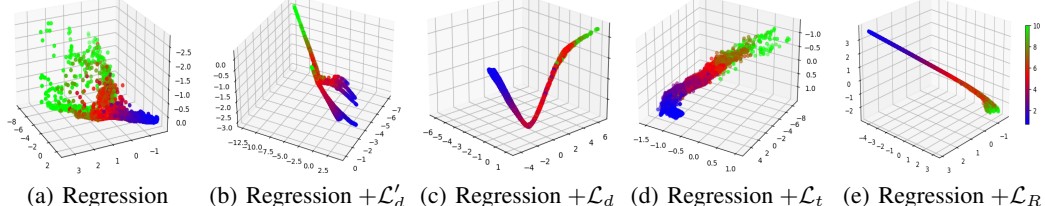

(a) Regression    (b) Regression $+\mathcal{L}'_d$  (c) Regression $+\mathcal{L}_d$  (d) Regression $+\mathcal{L}_t$  (e) Regression $+\mathcal{L}_R$

Figure 3: Visualization of the 3-dimensional feature space (we change the model' last hidden layer to dimension 3 for visualization) from the depth estimation task, based on a batch of 32 images. The target space is a 1-dimensional line. (b) $\mathcal{L}'_d$ encourages a lower intrinsic dimension yet fails to preserve the topology of the target space. (c) $\mathcal{L}_d$ takes the target space into consideration and can further preserve its topology. (d) $\mathcal{L}_t$ can enforce the topological similarity between the feature and target spaces. (e) Adding the $\mathcal{L}_t$ to $\mathcal{L}_d$ better preserves the topology of the target space.

and an illustration of its calculation is given in Figure 2(b). We define $E(\mathbf{S}) = \sum_{\gamma \in \mathrm{PH}_0(\mathrm{VR}(\mathbf{S}))} |I(\gamma)|$, where $|I(\gamma)|$ is the length of the interval $\gamma$.

## 4.2 PERSISTENT HOMOLOGY REGRESSION REGULARIZER

Birdal et al. (2021) suggests to estimate the intrinsic dimension as the slope between $\log E(\mathbf{Z}_n)$ and $\log n$, where $\mathbf{Z}_n$ is the set of $n$ samples from $\mathbf{Z}$. Let $\mathbf{e}' = [\log E(\mathbf{Z}_{n_1}), \log E(\mathbf{Z}_{n_2}), \cdots, \log E(\mathbf{Z}_{n_m})]$, where $\mathbf{Z}_{n_i}$ is the subset sampled from a batch, with size $n_i = |\mathbf{Z}_{n_i}|$. Let $n_i < n_j$ for $i < j$, and $\mathbf{n} = [\log n_1, \log n_2, \cdots, \log n_m]$. We can encourage a lower intrinsic dimension feature space by minimizing the slope between $\mathbf{e}'$ and $\mathbf{n}$, which can be estimated via the least square method:

$$\mathcal{L}'_d = (m \sum_{i=1}^{m} \mathbf{n}_i \mathbf{e}'_i - \sum_{i=1}^{m} \mathbf{n}_i \sum_{i=1}^{m} \mathbf{e}'_i)/(m \sum_{i=1}^{m} \mathbf{n}_i^2 - (\sum_{i=1}^{m} \mathbf{n}_i)^2). \tag{6}$$

While $\mathcal{L}'_d$ does encourage the feature space to have a lower intrinsic dimension, it fails to preserve the topology of the target space (see Figure 3(b)) and sometimes results in an intrinsic dimension even lower than the target space (see Figure 4, Swiss Roll, where the target space is 2 dimensional and the feature space is almost dimensional 1). As such, we opt to take the target space into consideration when minimizing the slope above. Let $\mathbf{Y}_n$ be the labels corresponding to $\mathbf{Z}_n$ and $\mathbf{e} = [\mathbf{e}_1, \mathbf{e}_2, \cdots, \mathbf{e}_m]$ where $\mathbf{e}_i = \log E(\mathbf{Z}_{n_i})/\log E(\mathbf{Y}_{n_i})$, we minimize slope between $\mathbf{e}$ and $\mathbf{n}$:

$$\mathcal{L}_d = |(m \sum_{i=1}^{m} \mathbf{n}_i \mathbf{e}_i - \sum_{i=1}^{m} \mathbf{n}_i \sum_{i=1}^{m} \mathbf{e}_i)/(m \sum_{i=1}^{m} \mathbf{n}_i^2 - (\sum_{i=1}^{m} \mathbf{n}_i)^2)|. \tag{7}$$

As shown in Figure 3(c) and Figure 4, $\mathcal{L}_d$ lowers the intrinsic dimension while better preserving the topology of the target space. Calculating $E_1(\mathbf{Z}_n)$ involves the 0-dimensional persistent homology $\mathrm{PH}_0(\mathrm{VR}(\mathbf{Z}_n))$. Specifically, calculating $\mathrm{PH}_0(\mathrm{VR}(\mathbf{Z}_n))$ turns out to find the minimum spanning tree of $\mathbf{Z}_n$ from its distance matrix $\mathbf{A}^{\mathbf{Z}_n}$, where $\mathbf{A}^{\mathbf{Z}_n}_{ij}$ is the Euclidean distance between $\mathbf{z}_i$ and $\mathbf{z}_j$. Calculating $E_1(\mathbf{Y}_n)$ is same. We denote $\pi^{\mathbf{Z}_n}, \pi^{\mathbf{Y}_n}$ the set of the index of edges in the minimum spanning trees of $\mathbf{Z}_n$ and $\mathbf{Y}_n$, respectively, and $\mathbf{A}^*[\pi^*]$ the corresponding length of the edges.

The topology autoencoder shows the topological similarity between the feature space and the target space can be enforced by preserving 0-dimensional topologically relevant distances from the target space and the label space. The topology part $\mathcal{L}_t$ is defined as:

$$\mathcal{L}_t = ||\mathbf{A}^{\mathbf{Z}_{n_m}}[\pi^{\mathbf{Z}_{n_m}}] - \mathbf{A}^{\mathbf{Y}_{n_m}}[\pi^{\mathbf{Z}_{n_m}}]||_2^2 + ||\mathbf{A}^{\mathbf{Z}_{n_m}}[\pi^{\mathbf{Y}_{n_m}}] - \mathbf{A}^{\mathbf{Y}_{n_m}}[\pi^{\mathbf{Y}_{n_m}}]||_2^2 \tag{8}$$

As shown in Figure 3(d) and Figure 4, $\mathcal{L}_t$ can preserve the topology of the target space, yet it fails to encourage a lower intrinsic dimension. We define the persistent homology regression regularizer, PH-Reg, as $\mathcal{L}_R = \mathcal{L}_d + \mathcal{L}_t$. As shown in Figure 3(e) and Figure 4, PH-Reg can both encourage a lower intrinsic dimension and preserve the topology of target space. We show our regression with PH-Reg (red dotted arrow) in Fig. 2(a). The final loss function $\mathcal{L}_{total}$ is defined as:

$$\mathcal{L}_{total} = \mathcal{L}_m + \lambda_t \mathcal{L}_t + \lambda_d \mathcal{L}_d, \tag{9}$$

Table 1: Results ($\mathcal{L}_{\mathrm{mse}}$) on the synthetic dataset. We report results as mean $\pm$ standard variance over 10 runs. **Bold** numbers indicate the best performance.

| Method | Swiss Roll | Mammoth | Torus | Circle |
|---|---|---|---|---|
| Baseline | $3.46 \pm 1.09$ | $201 \pm 72$ | $3.33 \pm 0.12$ | $0.175 \pm 0.004$ |
| $+\mathcal{L}'_d$ | $2.53 \pm 1.19$ | $195 \pm 57$ | $5.29 \pm 0.23$ | $0.157 \pm 0.027$ |
| $+\mathcal{L}_d$ | $1.14 \pm 0.63$ | $163 \pm 49$ | $1.47 \pm 0.07$ | $0.134 \pm 0.021$ |
| $+\mathcal{L}_t$ | $2.04 \pm 1.44$ | $60 \pm 63$ | $0.78 \pm 0.14$ | $0.040 \pm 0.009$ |
| $+\mathcal{L}_d + \mathcal{L}_t$ | $\mathbf{0.82 \pm 0.14}$ | $\mathbf{31 \pm 17}$ | $\mathbf{0.64 \pm 0.06}$ | $\mathbf{0.007 \pm 0.002}$ |

where $\mathcal{L}_m$ is the task-specific regression loss and $\lambda_d, \lambda_t$ are trade-off parameters.

## 5 EXPERIMENTS

We conduct experiments on four tasks: points coordinate prediction based on a synthetic dataset and three real-world regression tasks of depth estimation, super-resolution and age estimation. The target spaces of the three real-world regression tasks are topologically different, *i.e.*, a 1- dimensional line for depth estimation, 3-dimensional space for super-resolution and discrete points for age estimation.

### 5.1 COORDINATE PREDICTION ON THE SYNTHETIC DATASET

To verify the topological relationship between the feature space and target space, we synthetic a dataset that contains points sampled from topologically different objects, including swiss roll, torus, circle and the more complex object "mammoth" (Coenen & Pearce, 2019). We randomly sample 3000 points with coordinate $\mathbf{y} \in \mathbb{R}^3$ from each object, and the 3000 points are divided into 100 training points and 2900 testing points. Each point $\mathbf{y}_i$ is encoded into a 100 dimensional vector $\mathbf{x}_i = [f_1(\mathbf{y}_i), f_2(\mathbf{y}_i), f_3(\mathbf{y}_i), f_4(\mathbf{y}_i), \text{noise}]$, where the dimensions $1 - 4$ are signal and the rest 96 dimensions are noise. The coordinate prediction task aims to learn the mapping $G(\mathbf{x}) = \hat{\mathbf{y}}$ from $\mathbf{x}$ to $\mathbf{y}$, and the mean-squared error $\mathcal{L}_{\mathrm{mse}} = \frac{1}{N} \sum_i ||\hat{\mathbf{y}}_i - \mathbf{y}_i||_2^2$ is adopted as the evaluation metric. We use a two-layer fully connected neural network with 100 hidden units as the baseline architecture. The trade-off parameters $\lambda_d$ and $\lambda_t$ are default set to 10 and 100, respectively, while $\lambda_t$ is set to 10000 for Mammoth, 1 for Swiss Roll, and $\lambda_d$ is set to 1 for torus and circle. More details are in Appendix B.

Table 1 shows that encouraging a lower intrinsic dimension while considering the target space ($+\mathcal{L}_d$) enhances performance, particularly for Swiss Roll and Torus. In contrast, naively lowering the intrinsic dimension ($+\mathcal{L}'_d$) performs poorly and even worse than the baseline, *i.e.*, Tours. Enforcing the topology similarity between the feature space and target space($+\mathcal{L}'_t$) decreases the $\mathcal{L}_{\mathrm{mse}}$ by more than 70%, except for the Swiss roll. The best gains, however, are achieved by incorporating both $\mathcal{L}_t$ and $\mathcal{L}_d$, which decrease the $\mathcal{L}_{\mathrm{mse}}$ even up to 96% for the circle coordinate prediction task. Figure 4 shows some feature space visualization results based on t-SNE (100 dimensions $\rightarrow$ 3 dimensions). The feature space of the regression baseline shows a similar structure to the target space, especially for Swiss roll and mammoth, which indicates regression potentially captures the topology of the target space. Regression $+\mathcal{L}_t$ significantly preserves the topology of the target space. Regression $+\mathcal{L}_d$ potentially preserves the topology of the target space, *e.g.*, circle, while it primarily reduces the complexity of the feature space by maintaining the same intrinsic dimension as the target space. Combining both $\mathcal{L}_d$ and $\mathcal{L}_t$ in regression preserves the topology information while also reducing the complexity of the feature space, *i.e.*, lower its intrinsic dimension.

### 5.2 REAL-WORLD TASKS: DEPTH ESTIMATION, SUPER-RESOLUTION & AGE ESTIMATION

**Super-resolutuion on DIV2K dataset**: We exploit the DIV2K dataset (Timofte et al., 2017) for 4x super-resolution training (without the 2x pretrained model) and we evaluate on the validation set of DIV2K and the standard benchmarks: Set5 (Bevilacqua et al., 2012), Set14 (Zeyde et al., 2012), BSD100 (Martin et al., 2001), Urban100 (Huang et al., 2015). We follow the setting of Lim et al. (2017) and exploit their small-size EDSR model as our baseline architecture. We adopt the standard metric PNSR and trade-off parameters $\lambda_d$ and $\lambda_t$ are set to 0.1 and 1, respectively. Table 2 shows that both $\mathcal{L}_d$ and $\mathcal{L}_t$ contribute to improving the baseline and adding both terms has the largest impact.

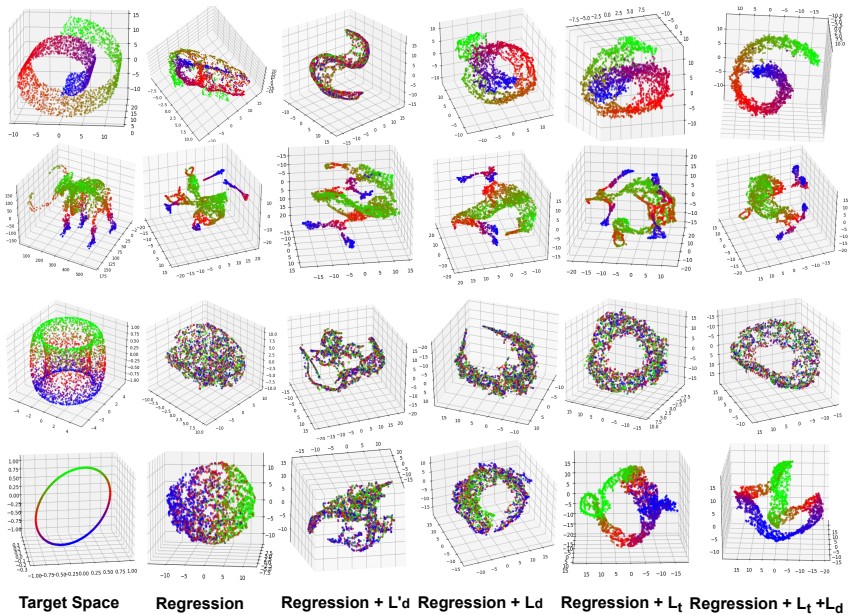

Figure 4: t-sne visualization of the 100-dimensional feature spaces with different target spaces.

Table 2: Quantitative comparison (PSNR(dB)) of super-resolution results with public benchmark and DIV2K validation set. **Bold** numbers indicate the best performance.

| Method | Set5 | Set14 | B100 | Urban100 | DIV2K |
|---|---|---|---|---|---|
| Baseline (Lim et al., 2017) | 32.241 | 28.614 | 27.598 | 26.083 | 28.997 |
| $+\mathcal{L}'_d$ | 32.252 | 28.625 | 27.599 | 26.078 | 28.989 |
| $+\mathcal{L}_d$ | 32.293 | 28.644 | 27.619 | 26.151 | 29.022 |
| $+\mathcal{L}_t$ | **32.322** | 28.673 | 27.624 | 26.169 | 29.031 |
| $+\mathcal{L}_d + \mathcal{L}_t$ | 32.288 | **28.686** | **27.627** | **26.179** | **29.038** |

Table 3: Results on AgeDB. **Bold** numbers indicate the best performance.

| Method | MAE ↓ | | | | GM ↓ | | | |
|---|---|---|---|---|---|---|---|---|
| | ALL | Many | Med. | Few | ALL | Many | Med. | Few |
| Baseline (Yang et al., 2021) | 7.77 | 6.62 | 9.55 | 13.67 | 5.05 | 4.23 | 7.01 | 10.75 |
| $+\mathcal{L}'_d$ | 7.81 | 6.96 | 8.88 | 12.91 | 4.95 | 4.45 | 5.54 | 9.91 |
| $+\mathcal{L}_d$ | 7.55 | 6.81 | **8.43** | **12.15** | 4.78 | 4.24 | **5.78** | **9.79** |
| $+\mathcal{L}_t$ | 7.50 | 6.58 | 8.79 | 12.67 | 4.84 | 4.22 | 6.12 | 9.12 |
| $+\mathcal{L}_d + \mathcal{L}_t$ | **7.48** | **6.52** | 8.71 | 13.19 | **4.74** | **4.06** | 6.17 | 9.77 |

**Age estimation on AgeDB-DIR dataset**: We exploit the AgeDB-DIR (Yang et al., 2021) for age estimation task. We follow the setting of Yang et al. (2021) and implement their regression baseline model, which uses ResNet-50 as a backbone. The evaluation metrics are MAE and geometric mean(GM), and the results are reported on the whole set and the three disjoint subsets, *i.e.*, Many, Med. and Few. The trade-off parameters $\lambda_d$ and $\lambda_t$ are set to $0.1$ and $1$, respectively. Table 3 shows that both $\mathcal{L}_t$ and $\mathcal{L}_d$ can achieve more than 0.2 overall improvements (*i.e.*, ALL) on both MAE and GM. Combining $\mathcal{L}_t$ and $\mathcal{L}_d$ can further boost the performance, and $\mathcal{L}'_d$ does not work.

**Depth estimation on NYU-Depth-v2 dataset**: We exploit the NYU-Depth-v2 (Silberman et al., 2012) for the depth estimation task. We follow the setting of Lee et al. (2019) and use ResNet50 (He et al., 2016) as our baseline architecture. We exploit the standard metrics of threshold accuracy $\delta_1, \delta_2, \delta_3$, average relative error (REL), root mean squared error (RMS) and average $\log_{10}$ error. The trade-off parameters $\lambda_d$ and $\lambda_t$ are both set to $0.1$. Table 4 shows that exploiting $\mathcal{L}_t$ and $\mathcal{L}_d$ results in reduction of $6.7\%$ and $8.9\%$ in the $\delta_1$ and $\delta_2$ errors, respectively.

Table 4: Depth estimation results with NYU-Depth-v2. **Bold** numbers indicate the best performance.

| Method | $\delta_1 \uparrow$ | $\delta_2 \uparrow$ | $\delta_3 \uparrow$ | REL $\downarrow$ | RMS $\downarrow$ | $\log_{10} \downarrow$ |
|---|---|---|---|---|---|---|
| Baseline (ResNet-50) | 0.792 | 0.955 | 0.990 | 0.153 | 0.512 | 0.064 |
| $+\mathcal{L}'_d$ | 0.804 | 0.954 | 0.988 | 0.151 | 0.502 | 0.063 |
| $+\mathcal{L}_d$ | 0.795 | **0.959** | **0.992** | 0.150 | 0.497 | 0.063 |
| $+\mathcal{L}_t$ | 0.798 | 0.958 | 0.990 | 0.149 | 0.502 | 0.063 |
| $+\mathcal{L}_d + \mathcal{L}_t$ | **0.807** | **0.959** | **0.992** | **0.147** | **0.494** | **0.062** |

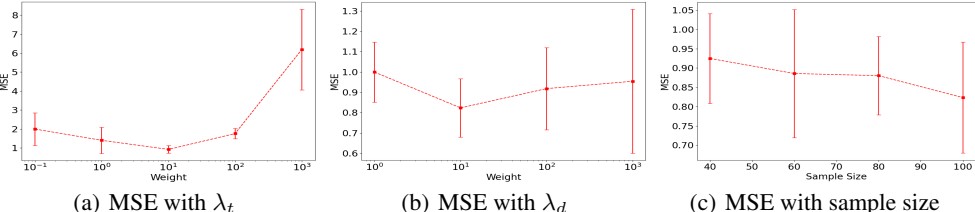

(a) MSE with $\lambda_t$      (b) MSE with $\lambda_d$      (c) MSE with sample size

Figure 5: Ablation study based on the swiss roll coordinate prediction.

Table 5: Quantitative comparison of the time consumption and memory usage on the synthetic dataset and NYU-Depth-v2, and the corresponding training times are 10000 and 1 epoch, respectively.

| $n_m$ | Regularizer | Coordinate Prediction (2 Layer MLP) | | Depth Estimation (ResNet-50) | |
|---|---|---|---|---|---|
| | | Training time (s) | Memory (MB) | Training time (s) | Memory (MB) |
| 0 | no regularizer | 8.88 | 959 | 1929 | 11821 |
| 100 | $\mathcal{L}_t$ | 175.06 | 959 | 1942 | 11833 |
| 100 | $\mathcal{L}_d$ | 439.68 | 973 | 1950 | 12211 |
| 100 | $\mathcal{L}_t + \mathcal{L}_d$ | 617.41 | 973 | 1980 | 12211 |
| 300 | $\mathcal{L}_t + \mathcal{L}_d$ | 956.97 | 1183 | 2370 | 12211 |

## 5.3 ABLATION STUDIES

**Hyperparameter $\lambda_t$ and $\lambda_d$**: We maintain $\lambda_d$ and $\lambda_t$ at their default value 10 for Swiss roll coordinate prediction, and we vary one of them to examine their impact. Figure 5(a) shows when $\lambda_t \leq 10$, the MSE decreases consistently as $\lambda_t$ increases. However, it tends to overtake the original learning objective when set too high, *i.e.*, 1000. Regarding the $\lambda_d$, as shown in Figure 5(b), MSE remains relatively stable over a large range of $\lambda_d$, with a slight increase in variance when $\lambda_d = 1000$.

**Sample Size ($n_m$)**: In practice, we model the feature space using a limited number of samples within a batch. For dense prediction tasks, the available No. of samples is very large (No. pixels per image $\times$ batch size), while it is constrained to the batch size for image-wise prediction tasks. We investigate the influence of $n_m$ from Eq. 7 and 8 on Swiss roll coordinate prediction. Figure 5(c) shows our PH-Reg performs better with a larger $n_m$, while maintaining stability even with a small $n_m$.

**Efficiency**: Efficiency-wise, the computing complexity equals finding the minimum spanning tree from the distance matrix of the samples, which have a complexity of $\mathcal{O}(n_m^2 \log n_m)$ using the simple Kruskal's Algorithm, and it can speed up with some advanced methods (Bauer, 2021). The synthetic experiments (Table 5) use a simple 2-layer MLP, so the regularizer adds significant computing time. However, the real-world experiments on depth estimation (Table 5) use a ResNet-50 backbone, and the added time and memory are negligible (18.6% and 0.3%, respectively), even with $n_m = 300$. Note that these increases are only during training and do not add computing demands for inference.

## 6 CONCLUSION

In this paper, we establish novel connections between topology and the IB principle for regression representation learning. The established connections imply that the desired **Z** should exhibit topological similarity to the target space and share the same intrinsic dimension as the target space. Inspired by the connections, we proposed a regularizer to learn the desired **Z**. Experiments on synthetic and real-world regression tasks demonstrate its benefits.

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

# A PROOFS

## A.1 PROOF OF THE THEOREM 1

**Theorem 1** *Optimizing the Information Bottleneck trade-off Lagrangian is equivalent to minimizing the conditional entropies $\mathcal{H}(\mathbf{Y}|\mathbf{Z})$ and $\mathcal{H}(\mathbf{Z}|\mathbf{Y})$.*

**Proof** Optimizing the information bottleneck trade-off Lagrangian : $\min_{\mathbf{Z}}\{I(\mathbf{Z};\mathbf{X}) - \beta I(\mathbf{Z};\mathbf{Y})\}$ can be written as $\min_{\mathbf{Z}} \frac{\mathcal{I}(\mathbf{Z};\mathbf{X})}{\beta\mathcal{I}(\mathbf{Z};\mathbf{Y})}$, we note:

$$\frac{\mathcal{I}(\mathbf{Z};\mathbf{X})}{\beta\mathcal{I}(\mathbf{Z};\mathbf{Y})} = \frac{\mathcal{H}(\mathbf{Z}) - \mathcal{H}(\mathbf{Z}|\mathbf{X})}{\beta\mathcal{I}(\mathbf{Z};\mathbf{Y})} \tag{10}$$

$$= \frac{\mathcal{I}(\mathbf{Z};\mathbf{Y}) + \mathcal{H}(\mathbf{Z}|\mathbf{Y}) - \mathcal{H}(\mathbf{Z}|\mathbf{X})}{\beta\mathcal{I}(\mathbf{Z};\mathbf{Y})} \tag{11}$$

$$= \frac{1}{\beta} + \frac{\mathcal{H}(\mathbf{Z}|\mathbf{Y}) - \mathcal{H}(\mathbf{Z}|\mathbf{X})}{\beta(\mathcal{H}(\mathbf{Y}) - \mathcal{H}(\mathbf{Y}|\mathbf{Z}))}. \tag{12}$$

Since the mapping from $\mathbf{X}$ to $\mathbf{Z}$ is deterministic, $\mathcal{H}(\mathbf{Z}|\mathbf{X})$ is canceled to 0, based on the above equation, we have:

$$\min_{\mathbf{Z}} \frac{\mathcal{I}(\mathbf{Z};\mathbf{X})}{\beta\mathcal{I}(\mathbf{Z};\mathbf{Y})} = \min_{\mathbf{Z}} \frac{\mathcal{H}(\mathbf{Z}|\mathbf{Y}) - \mathcal{H}(\mathbf{Z}|\mathbf{X})}{\beta(\mathcal{H}(\mathbf{Y}) - \mathcal{H}(\mathbf{Y}|\mathbf{Z}))} = \min_{\mathbf{Z}} \frac{\mathcal{H}(\mathbf{Z}|\mathbf{Y})}{\beta(\mathcal{H}(\mathbf{Y}) - \mathcal{H}(\mathbf{Y}|\mathbf{Z}))}. \tag{13}$$

Since $\mathcal{H}(\mathbf{Y})$ is a constant and $\mathcal{H}(\mathbf{Y}|\mathbf{Z}) < \mathcal{H}(\mathbf{Y})$, minimizing $\frac{\mathcal{I}(\mathbf{Z};\mathbf{X})}{\beta\mathcal{I}(\mathbf{Z};\mathbf{Y})}$ equals minimizing $\mathcal{H}(\mathbf{Y}|\mathbf{Z})$ and $\mathcal{H}(\mathbf{Z}|\mathbf{Y})$. □

## A.2 PROOF OF THE THEOREM 2 AND PROPOSITION 1

**Theorem 2** *We are given dataset $S = \{\mathbf{x}_i, \mathbf{z}_i, \mathbf{y}_i\}_{i=1}^N$ sampled from distribution $P$, where $\mathbf{x}_i$ is the input, $\mathbf{z}_i$ is the corresponding representation, and $\mathbf{y}_i$ is the label. Let $d_{max} = \max_{\mathbf{y}\in\mathcal{Y}} \min_{\mathbf{y}_i\in S} ||\mathbf{y} - \mathbf{y}_i||_2$ be the maximum distance of $\mathbf{y}$ to its nearset $\mathbf{y}_i$. Assume $(\mathbf{Z}|\mathbf{Y} = \mathbf{y}_i)$ follows a disribution $\mathcal{D}$ and the following holds:*

$$\mathbb{E}_{\mathbf{z}\sim\mathcal{D}}[||\mathbf{z} - \bar{\mathbf{z}}||_2] \leq Q(\mathcal{H}(\mathcal{D})), \tag{14}$$

*where $\bar{\mathbf{z}}$ is the mean of the distribution $\mathcal{D}$ and $Q(\mathcal{H}(\mathcal{D}))$ is some function of $\mathcal{H}(\mathcal{D})$. The above implies the dispersion of the distribution $\mathcal{D}$ is bounded by its entropy, which usually is the case, like the multivariate normal distribution and the uniform distribution. Assume the regressor $f$ is $L_1$-Lipschitz continuous, then as $d_{max} \to 0$, we have*

$$\mathbb{E}_{\{\mathbf{x},\mathbf{z},\mathbf{y}\}\sim P}[||f(\mathbf{z}) - \mathbf{y}||_2] \leq \mathbb{E}_{\{\mathbf{x},\mathbf{z},\mathbf{y}\}\sim S}(||f(\mathbf{z}) - \mathbf{y}||_2) + 2L_1 Q(\mathcal{H}(\mathbf{Z}|\mathbf{Y})) \tag{15}$$

**Proof** For any sample $\{\mathbf{x}_i, \mathbf{z}_i, \mathbf{y}_i\}$, we define its local neighborhood set $N_i$ as

$$N_i = \{\{\mathbf{x},\mathbf{z},\mathbf{y}\} : ||\mathbf{y} - \mathbf{y}_i||_2 < ||\mathbf{y} - \mathbf{y}_j||_2, j \neq i, p(\mathbf{y}) > 0\}. \tag{16}$$

For each set $N_i$, we have

$$\mathbb{E}_{\{\mathbf{x},\mathbf{z},\mathbf{y}\}\sim N_i}[||f(\mathbf{z}) - \mathbf{y}||_2] = \mathbb{E}_{\{\mathbf{x},\mathbf{z},\mathbf{y}\}\sim N_i}[||f(\mathbf{z}) - f(\mathbf{z}_i) + f(\mathbf{z}_i) - \mathbf{y}_i + \mathbf{y}_i - \mathbf{y}||_2] \tag{17}$$

$$\leq \mathbb{E}_{\{\mathbf{x},\mathbf{z},\mathbf{y}\}\sim N_i}[||f(\mathbf{z}) - f(\mathbf{z}_i)||_2] + \mathbb{E}_{\{\mathbf{x},\mathbf{z},\mathbf{y}\}\sim N_i}[||f(\mathbf{z}_i) - \mathbf{y}_i||_2] + \mathbb{E}_{\{\mathbf{x},\mathbf{z},\mathbf{y}\}\sim N_i}[||\mathbf{y}_i - \mathbf{y}||_2] \tag{18}$$

$$\leq L_1 \mathbb{E}_{\{\mathbf{x},\mathbf{z},\mathbf{y}\}\sim N_i}[||\mathbf{z} - \mathbf{z}_i||_2] + \mathbb{E}_{\{\mathbf{x},\mathbf{z},\mathbf{y}\}\sim N_i}[||f(\mathbf{z}_i) - \mathbf{y}_i||_2] + d_{max} \tag{19}$$

$$= L_1 \mathbb{E}_{\{\mathbf{x},\mathbf{z},\mathbf{y}\}\sim N_i}[||\mathbf{z} - \bar{\mathbf{z}}_i + \bar{\mathbf{z}}_i - \mathbf{z}_i||_2] + \mathbb{E}_{\{\mathbf{x},\mathbf{z},\mathbf{y}\}\sim N_i}[||f(\mathbf{z}_i) - \mathbf{y}_i||_2] + d_{max} \tag{20}$$

$$\leq L_1 \mathbb{E}_{\{\mathbf{x},\mathbf{z},\mathbf{y}\}\sim N_i}[||\mathbf{z} - \bar{\mathbf{z}}_i||_2 + ||\bar{\mathbf{z}}_i - \mathbf{z}_i||_2] + \mathbb{E}_{\{\mathbf{x},\mathbf{z},\mathbf{y}\}\sim N_i}[||f(\mathbf{z}_i) - \mathbf{y}_i||_2] + d_{max} \tag{21}$$

$$= L_1 \mathbb{E}_{\{\mathbf{x},\mathbf{z},\mathbf{y}\}\sim N_i}[||\mathbf{z} - \bar{\mathbf{z}}_i||_2] + L_1 ||\bar{\mathbf{z}}_i - \mathbf{z}_i||_2 + \mathbb{E}_{\{\mathbf{x},\mathbf{z},\mathbf{y}\}\sim N_i}[||f(\mathbf{z}_i) - \mathbf{y}_i||_2] + d_{max} \tag{22}$$

We denote the probability distribution over $\{N_i\}$ as $P'$, where $P(N_i) = P(\{\mathbf{x},\mathbf{z},\mathbf{y}\} \in N_i)$. Then, we have

$$\mathbb{E}_{\{\mathbf{x},\mathbf{z},\mathbf{y}\}\sim P}[||f(\mathbf{z}) - \mathbf{y}||_2] = \mathbb{E}_{N_i\sim P'}\mathbb{E}_{\{\mathbf{x},\mathbf{z},\mathbf{y}\}\sim N_i}[||f(\mathbf{z}) - \mathbf{y}||_2] \tag{23}$$

$$\leq \mathbb{E}_{N_i\sim P'}[L_1 \mathbb{E}_{\{\mathbf{x},\mathbf{z},\mathbf{y}\}\sim N_i}[||\mathbf{z} - \bar{\mathbf{z}}_i||_2] + L_1 ||\bar{\mathbf{z}}_i - \mathbf{z}_i||_2 + \mathbb{E}_{\{\mathbf{x},\mathbf{z},\mathbf{y}\}\sim N_i}[||f(\mathbf{z}_i) - \mathbf{y}_i||_2] + d_{max}] \tag{24}$$

$$= L_1 \mathbb{E}_{N_i\sim P'}\mathbb{E}_{\{\mathbf{x},\mathbf{z},\mathbf{y}\}\sim N_i}[||\mathbf{z} - \bar{\mathbf{z}}_i||_2] + L_1 \mathbb{E}_{N_i\sim P'}||\bar{\mathbf{z}}_i - \mathbf{z}_i||_2 + \mathbb{E}_{\{\mathbf{x},\mathbf{z},\mathbf{y}\}\sim S}(||f(\mathbf{z}_i) - \mathbf{y}_i||_2) + d_{max} \tag{25}$$

As $d_{max} \to 0$, we can approximate $\mathbb{E}_{N_i \sim P'} \mathbb{E}_{\{\mathbf{x}, \mathbf{z}, \mathbf{y}\} \sim N_i} [||\mathbf{z} - \bar{\mathbf{z}}_i||_2]$ as $\mathbb{E}_{\mathbf{y}_i \sim \mathcal{Y}} \mathbb{E}_{\{(\mathbf{x}, \mathbf{z}, \mathbf{y}) | \mathbf{y} = \mathbf{y}_i\}} [||\mathbf{z} - \bar{\mathbf{z}}_i||_2]$. Since $(\mathbf{Z} | \mathbf{Y} = \mathbf{y}_i) \sim \mathcal{D}$, we have $\mathcal{H}(\mathbf{Z} | \mathbf{Y}) = \mathbb{E}_{y \sim \mathcal{Y}} \mathcal{H}(\mathbf{Z} | \mathbf{Y} = y) = \mathcal{H}(\mathbf{Z} | \mathbf{Y} = \mathbf{y}_i) = \mathcal{H}(\mathbf{Z} | \mathbf{Y} = \mathbf{y}_j) = \mathcal{H}(\mathcal{D})$ for all $1 \le i, j \le N$, and $\mathbb{E}_{N_i \sim P'} ||\mathbf{z}_i - \bar{\mathbf{z}}_i||_2$ can thus be approximate as $\mathbb{E}_{\{(\mathbf{x}, \mathbf{z}, \mathbf{y}) | \mathbf{y} = \mathbf{y}_i\}} ||\mathbf{z} - \bar{\mathbf{z}}_i||_2$. We have:

$$\mathbb{E}_{\{\mathbf{x}, \mathbf{z}, \mathbf{y}\} \sim P} [||f(\mathbf{z}) - \mathbf{y}||_2] \tag{26}$$

$$\le L_1 \mathbb{E}_{N_i \sim P'} \mathbb{E}_{\{\mathbf{x}, \mathbf{z}, \mathbf{y}\} \sim N_i} [||\mathbf{z} - \bar{\mathbf{z}}_i||_2] + L_1 \mathbb{E}_{N_i \sim P'} ||\bar{\mathbf{z}}_i - \mathbf{z}_i||_2 + \mathbb{E}_{\{\mathbf{x}, \mathbf{z}, \mathbf{y}\} \sim S} (||f(\mathbf{z}_i) - \mathbf{y}_i||_2) + d_{max} \tag{27}$$

$$= L_1 \mathbb{E}_{y_i \sim \mathcal{Y}} \mathbb{E}_{\{(\mathbf{x}, \mathbf{z}, y) | y = y_i\}} [||\mathbf{z} - \bar{\mathbf{z}}_i||_2] + L_1 \mathbb{E}_{\{(\mathbf{x}, \mathbf{z}, y) | y = y_i\}} ||\mathbf{z}_i - \bar{\mathbf{z}}_i||_2 + \mathbb{E}_{\{\mathbf{x}, \mathbf{z}, \mathbf{y}\} \sim S} (||f(\mathbf{z}_i) - \mathbf{y}_i||_2) \tag{28}$$

$$\le L_1 \mathbb{E}_{y_i \sim \mathcal{Y}} [Q(\mathcal{H}(\mathbf{Z} | \mathbf{Y} = \mathbf{y}_i))] + L_1 Q(\mathcal{H}(\mathbf{Z} | \mathbf{Y} = \mathbf{y}_i)) + \mathbb{E}_{\{\mathbf{x}, \mathbf{z}, \mathbf{y}\} \sim S} (||f(\mathbf{z}_i) - \mathbf{y}_i||_2) \tag{29}$$

$$= 2 L_1 Q(\mathcal{H}(\mathbf{Z} | \mathbf{Y})) + \mathbb{E}_{\{\mathbf{x}, \mathbf{z}, \mathbf{y}\} \sim S} (||f(\mathbf{z}_i) - \mathbf{y}_i||_2) \tag{30}$$

$\square$

**Proposition 1** *If $\mathcal{D}$ is a multivariate normal distribution $\mathcal{N}(\bar{\mathbf{z}}, \Sigma = k\mathbf{I})$, where $k > 0$ is a scalar and $\bar{\mathbf{z}}$ is the mean of the distribution $\mathcal{D}$. Then, the function $Q(\mathcal{H}(\mathcal{D}))$ in Theorem 2 can be selected as $Q(\mathcal{H}(\mathcal{D})) = \sqrt{\frac{d(e^{2\mathcal{H}(\mathcal{D})})^{\frac{1}{d}}}{2\pi e}}$, where $d$ is the dimension of $\mathbf{z}$. If $\mathcal{D}$ is a uniform distribution, then the $Q(\mathcal{H}(\mathcal{D}))$ can be selected as $Q(\mathcal{H}(\mathcal{D})) = \frac{e^{\mathcal{H}(\mathcal{D})}}{\sqrt{12}}$.*

**Proof** We first consider the case when $\mathcal{D} \sim \mathcal{N}(\bar{\mathbf{z}}, \Sigma = k\mathbf{I})$. Assume $\mathbf{Z} \sim \mathcal{N}(\bar{\mathbf{z}}, \Sigma)$, then $\mathcal{H}(\mathbf{Z}) = \frac{1}{2} \log(2\pi e)^n |\Sigma|$:

$$\mathcal{H}(\mathbf{Z}) = - \int_{\mathbf{z}} p(\mathbf{z}) \log(p(\mathbf{z})) d\mathbf{z} \tag{31}$$

$$= - \int_{\mathbf{z}} p(\mathbf{z}) \log \frac{1}{(\sqrt{2\pi})^d |\Sigma|^{\frac{1}{2}}} e^{\frac{-1}{2}(\mathbf{z} - \bar{\mathbf{z}})^\top \Sigma^{-1} (\mathbf{z} - \bar{\mathbf{z}})} d\mathbf{z} \tag{32}$$

$$= - \int_{\mathbf{z}} p(\mathbf{z}) \log \frac{1}{(\sqrt{2\pi})^d |\Sigma|^{\frac{1}{2}}} d\mathbf{z} - \int_{\mathbf{z}} p(\mathbf{z}) \log e^{\frac{1}{2}(\mathbf{z} - \bar{\mathbf{z}})^\top \Sigma^{-1} (\mathbf{z} - \bar{\mathbf{z}})} d\mathbf{z} \tag{33}$$

$$= \frac{1}{2} \log(2\pi)^d |\Sigma| + \frac{\log e}{2} \mathbb{E}[\sum_{i,j} (\mathbf{z}_i - \bar{\mathbf{z}}_i)(\Sigma^{-1})_{ij}(\mathbf{z}_j - \bar{\mathbf{z}}_j)] \tag{34}$$

$$= \frac{1}{2} \log(2\pi)^d |\Sigma| + \frac{\log e}{2} \mathbb{E}[\sum_{i,j} (\mathbf{z}_i - \bar{\mathbf{z}}_i)(\mathbf{z}_j - \bar{\mathbf{z}}_j)(\Sigma^{-1})_{ij}] \tag{35}$$

$$= \frac{1}{2} \log(2\pi)^d |\Sigma| + \frac{\log e}{2} \sum_{i,j} \mathbb{E}[(\mathbf{z}_i - \bar{\mathbf{z}}_i)(\mathbf{z}_j - \bar{\mathbf{z}}_j)](\Sigma^{-1})_{ij} \tag{36}$$

$$= \frac{1}{2} \log(2\pi)^d |\Sigma| + \frac{\log e}{2} \sum_j \sum_i \Sigma_{ji}(\Sigma^{-1})_{ij} \tag{37}$$

$$= \frac{1}{2} \log(2\pi)^d |\Sigma| + \frac{\log e}{2} \sum_j (\Sigma \Sigma^{-1})_j \tag{38}$$

$$= \frac{1}{2} \log(2\pi)^d |\Sigma| + \frac{\log e}{2} \sum_j I_{jj} \tag{39}$$

$$= \frac{1}{2} \log(2\pi)^d |\Sigma| + \frac{\log e}{2} \tag{40}$$

$$= \frac{1}{2} \log(2\pi e)^d |\Sigma| \tag{41}$$

We have the following:

$$\mathbb{E}[||\mathbf{z} - \bar{\mathbf{z}}||_2^2] = \text{tr}(\Sigma) = dk. \tag{42}$$

The following also holds:

$$|\Sigma| = k^d. \tag{43}$$

Thus, we have:

$$(\mathbb{E}[||\mathbf{z} - \bar{\mathbf{z}}||_2])^2 \le \mathbb{E}[||\mathbf{z} - \bar{\mathbf{z}}||_2^2] = d|\Sigma|^{\frac{1}{d}} = d\left(\frac{e^{2\mathcal{H}(\mathbf{Z})}}{(2\pi e)^d}\right)^{\frac{1}{d}} = \frac{d(e^{2\mathcal{H}(\mathbf{Z})})^{\frac{1}{d}}}{2\pi e} \tag{44}$$

Finally,

$$\mathbb{E}[||\mathbf{z} - \bar{\mathbf{z}}||_2] \le \sqrt{\frac{d(e^{2\mathcal{H}(\mathbf{Z})})^{\frac{1}{d}}}{2\pi e}} \tag{45}$$

Thus, $Q(\mathcal{H}(\mathcal{D}))$ in Theorem 2 can be selected as $Q(\mathcal{H}(\mathcal{D})) = \sqrt{\frac{d(e^{2\mathcal{H}(\mathcal{D})})^{\frac{1}{d}}}{2\pi e}}$, when $\mathcal{D} \sim \mathcal{N}(\bar{\mathbf{z}}, \Sigma = k\mathbf{I})$.

Similarly, if $\mathcal{D}$ is a uniform distribution $U(a, b)$, then its variance is given by:

$$\mathbb{E}[||\mathbf{z} - \bar{\mathbf{z}}||_2^2] = \frac{(b - a)^2}{12}, \tag{46}$$

and its entropy is given by:

$$\mathcal{H}(\mathbf{D}) = \log(b - a). \tag{47}$$

We have:

$$(\mathbb{E}[||\mathbf{z} - \bar{\mathbf{z}}||_2])^2 \le \mathbb{E}[||\mathbf{z} - \bar{\mathbf{z}}||_2^2] = \frac{(b - a)^2}{12} = \frac{e^{2\mathcal{H}(\mathcal{D})}}{12} \tag{48}$$

Finally,

$$\mathbb{E}[||\mathbf{z} - \bar{\mathbf{z}}||_2] \le \frac{e^{\mathcal{H}(\mathcal{D})}}{\sqrt{12}} \tag{49}$$

Thus, $Q(\mathcal{H}(\mathcal{D}))$ in Theorem 2 can be selected as $Q(\mathcal{H}(\mathcal{D})) = \frac{e^{\mathcal{H}(\mathcal{D})}}{\sqrt{12}}$, when $\mathcal{D}$ is a uniform distribution

$\square$

## A.3 PROOF OF THE THEOREM 3

**Theorem 3** *Assume that $\mathbf{z}$ lies in a manifold $\mathcal{M}$ and the $\mathcal{M}_i \subset \mathcal{M}$ is a manifold corresponding to the distribution $(\mathbf{z}|\mathbf{y} = \mathbf{y}_i)$. Assume for all features $\mathbf{z}_i \in \mathcal{M}_i$, the following holds:*

$$\int_{||\mathbf{z} - \mathbf{z}_i|| \le \epsilon} P(\mathbf{z}) d\mathbf{z} = C(\epsilon), \tag{50}$$

*where $C(\epsilon)$ is some function of $\epsilon$. The above imposes a constraint where the distribution $(\mathbf{z}|\mathbf{y} = \mathbf{y}_i)$ is uniformly distributed across $\mathcal{M}_i$. Then, as $\epsilon \to 0^+$, we have:*

$$\mathcal{H}(\mathbf{Z}|\mathbf{Y}) = \mathbb{E}_{\mathbf{y}_i \sim \mathcal{Y}} \mathcal{H}(\mathbf{Z}|\mathbf{Y} = \mathbf{y}_i) = \mathbb{E}_{\mathbf{y}_i \sim \mathcal{Y}}[-\log(\epsilon) Dim_{ID} \mathcal{M}_i + \log \frac{K}{C(\epsilon)}], \tag{51}$$

*for some fixed scalar K. $Dim_{ID} \mathcal{M}_i$ is the intrinsic dimension of the manifold $\mathcal{M}_i$.*

**Proof** By using the same proof technique as [Ghosh & Motani (2023), Proposition 1], we can show

$$\mathcal{H}(\mathbf{Z}|\mathbf{Y} = \mathbf{y}_i) = -\log(\epsilon) Dim_{ID} \mathcal{M}_i + \log \frac{K}{C(\epsilon)}, \tag{52}$$

Since $\mathcal{H}(\mathbf{Z}|\mathbf{Y}) = \mathbb{E}_{\mathbf{y}_i \sim \mathcal{Y}} \mathcal{H}(\mathbf{Z}|\mathbf{Y} = \mathbf{y}_i)$, the result follows. $\square$

### A.4 PROOF OF THE PROPOSITION 2

**Proposition 2** *If the representation $\mathbf{Z}$ is optimal and the mapping $f'$ between $\mathbf{Z}$ and $\mathbf{Y}$ and its inverse $f'^{-1}$ are continuous, then $\mathbf{Z}$ is homeomorphic to $\mathbf{Y}$.*

**Proof** If $\mathbf{Z}$ is optimal, then we have $\mathcal{H}(\mathbf{Y}|\mathbf{Z}) = 0$. Thus, for each $\mathbf{z}_i \in \mathbf{Z}$, there exists and only exists one $\mathbf{y}_i$ corresponding to the $\mathbf{z}_i$, and thus the mapping function $f$ exists. $\mathbf{Z}$ is optimal also means $\mathcal{H}(\mathbf{Z}|\mathbf{Y}) = 0$, and thus for each $\mathbf{y}_i$, there exist and only exist one $\mathbf{z}_i$ corresponding to the $\mathbf{y}_i$. Thus, the mapping function $f'$ is a bijection, and since $f'$ and $f'^{-1}$ are continuous, $\mathbf{Z}$ is homeomorphic to $\mathbf{Y}$. $\qquad\square$

## B DETAILS ABOUT THE SYNTHETIC DATASET

We encode coordinates $\mathbf{y} \in \mathbb{R}^3$ into 100 dimensional vectors $\mathbf{x}_i = [f_1(\mathbf{y}_i), f_2(\mathbf{y}_i), f_3(\mathbf{y}_i), f_4(\mathbf{y}_i), \text{noise}]$, where the dimensions $1-4$ are signal and the rest 96 dimensions are noise. The encoder functions $f_i$ are defined as:

- $f_1(\mathbf{y}_i) = y_{i_1} + y_{i_2} + y_{i_3}$
- $f_2(\mathbf{y}_i) = y_{i_1} + y_{i_2} - y_{i_3}$
- $f_3(\mathbf{y}_i) = y_{i_1} - y_{i_2} + y_{i_3}$
- $f_4(\mathbf{y}_i) = -y_{i_1} + y_{i_2} + y_{i_3}$

As shown above, the accurate coordinates $\mathbf{y}_i$ can be obtained correctly when $f_1(\mathbf{y}_i), f_2(\mathbf{y}_i), f_3(\mathbf{y}_i), f_4(\mathbf{y}_i)$ are given. We introduce noise to the remaining 96 dimensions by using $f_1, f_2, f_3, f_4$ on other randomly selected samples $\mathbf{y}_j$. The proximity of $\mathbf{y}_j$ to $\mathbf{y}_i$ can be intuitively seen as an indicator of the noise's relationship to the signal.

## C EVALUATION METRICS

Definition of the evaluation metrics for depth estimation and age estimation are given below.

**Depth Estimation.** We denote the predicted depth at position $p$ as $y_p$ and the corresponding ground truth depth as $y'_p$, the total number of pixels is $n$. The metrics are: 1) threshold accuracy $\delta_1 \triangleq \%$ of $y_p$, s.t. $\max(\frac{y_p}{y'_p}, \frac{y'_p}{y_p}) < t_1$, where $t_1 = 1.25$; 2) average relative error (REL): $\frac{1}{n} \sum_p \frac{|y_p - y'_p|}{y_p}$; 3) root mean squared error (RMS): $\sqrt{\frac{1}{n} \sum_p (y_p - y'_p)^2}$; 4) average ($\log_{10}$ error): $\frac{1}{n} \sum_p |\log_{10}(y_p) - \log_{10}(y'_p)|$.

**Age Estimation.** Given $N$ images for testing, $y_i$ and $y'_i$ are the $i$-th prediction and ground-truth, respectively. The evaluation metrics include 1)MAE: $\frac{1}{N} \sum_{i=1}^{N} |y_i - y'_i|$, and 2)Geometric Mean (GM): $(\prod_{i=1}^{N} |y_i - y'_i|)^{\frac{1}{N}}$.

