# OpenReview forum: "Deep Regression Representation Learning with Topology"
_ICLR.cc/2024/Conference — ICLR 2024 Conference Withdrawn Submission_

### Official Review · Reviewer_iGsx · 2023-10-31

**Soundness:** 2 fair
**Presentation:** 2 fair
**Contribution:** 2 fair
**Rating:** 3
**Confidence:** 3

**Summary:**

The paper uses the task of regression to study connections between the Information Bottleneck principle and the topology of feature space. Based on these connections, the authors introduce a regularisation scheme

**Strengths:**

The paper introduces an interesting approach to study representation learning from the point of view of the information bottleneck principle, but there are some problems in the presentation that prevent me from giving a fair evaluation of the analysis (see below).

Additionally, not being an expert in topology, I am not able to assess the soundness and value of the method introduced in section 5.

**Weaknesses:**

There are some points of the theoretical analysis that I do not understand and prevent me from giving a proper evaluation. The analysis hinges on theorem 1, but there are two aspects of it that I do not understand:

1. Minimising the difference is not generally equivalent to minimising the ratio, implying that the equivalence is a result of I being the mutual information. Can the author provide details on this equivalence? Is there a difference between I and \mathcal{I} in the proof of theorem 10?

2. Both Z and Y are deterministic functions of X. What does it imply for the entropy of Z conditioned on Y? Is it only meaningful when the mapping from X to Y is one-to-many? Does your theory suffer from the problems associated with having deterministic functions, as discussed in 'ON THE INFORMATION BOTTLENECK THEORY OF DEEP LEARNING' by Saxe et al, ICLR 2018?

Additionally,

-The accessibility of Section 4 is limited to readers with advanced knowledge in topology.

-The experimental evaluation is limited to a very simple architecture (two-layer network with 100 hidden units)

**Questions:**

See weaknesses above. In addition:

1. What is the meaning of `regression representation learning'? Do you mean representation learning in a regression task?

2. Are Swiss Roll and Mammoth standard jargon in topology?

---

> ### Author Response · Authors · 2023-11-22
>
> Dear Reviewer iGsx,
>
> Thank you for your insightful and detailed comments.
>
> For the weaknesses:
>
> **W1. Minimizing the difference is not equivalent to minimizing the ratio:** Thanks for pointing this out. I and \mathcal{I} both represent mutual information; we will revise the typos and theorem 1.
>
> **W2. Deterministic functions of $X$:**  There is a misunderstanding between the predicted $Y’$ by neural networks and the target $Y$. The predicted $Y’$ is a deterministic function of $X$, while $Y$ is not a deterministic function of $X$. Our theorem holds for many-to-one mapping from $X$ to $Y$. We will extend it to many-to-many as suggested by reviewer awEv. Saxe et al. discussed some problems associated with deterministic mapping when estimating mutual information. To estimate the mutual information, they suggest analyzing a new variable $Z’$ with additive noise, let $Z’ = Z + N$, where $N$ is the noise. The deterministic mapping assumption in theorem 1 is to cancel the term $\mathcal H(Z|X)$. If N follows some fixed distribution, then $\mathcal H(Z’|X) = \mathcal H(N)$ is a constant and thus also can be canceled.
>
>
> **Section 4 is limited to readers with advanced knowledge in topology:** Thanks for pointing this out. We will improve the readability.
>
> **Limited to a very simple architecture**: Experiments on real-world tasks using some complex architectures, like the EDSR model on DIV2K.
>
> **Meaning of `regression representation learning’'**: Yes, it means representation learning in regression tasks.
>
> **Are Swiss Roll and Mammoth standard jargon in topology?**: Swiss Roll is widely exploited and we follow RTD [arxiv 2302.00136] to exploit Mammoth.

---

### Official Review · Reviewer_awEv · 2023-10-31

**Soundness:** 2 fair
**Presentation:** 2 fair
**Contribution:** 3 good
**Rating:** 5
**Confidence:** 3

**Summary:**

The paper introduces Persistent Homology Regression Regularizer (PH-Reg) as a novel regularization strategy inspired by the Information Bottleneck (IB) principle with the goal of regularizing intermediate representation to preserve the topological structure of the (regression) targets while reducing the intrinsic dimensionality. A theoretical section relates the intrinsic dimensionality of the representation to the IB principle and generalization error, justifying the need for PH-Reg. An experimental section validates the effectiveness of the proposed method by analyzing the effect of the regularization components that encourage lower intrinsic dimension $\mathcal{L}_d$ and preserve the topology of the target space $\mathcal{L}_t$ on both synthetic and real-world tasks.

**Strengths:**

1) The paper draws interesting and novel connections between the Information Bottleneck principle and the topology and intrinsic dimensions of the representations with respect to the targets


2) The experimental section covers a significant number of diverse datasets, integrating quantitative results with qualitative visualizations


3) The authors include an estimation of the additional computational and memory cost of Ph-Reg demonstrating that the overhead of the proposed method is small compared to the cost of training large architectures.

**Weaknesses:**

## Main Concerns

1) **Clarity**
   1) The underlying assumptions used to prove the statements in Section 3 are not fully clarified. Theorem 1 implicitly restricts the considered representations to deterministic functions of $\bf x$. Previous literature [1] has shown the benefit of using stochastic encoders, for which the result from Theorem 1 is not applicable.
   2) Theorem 3 assumes a uniform (conditional) distribution on the manifold $\mathcal{M}_i$ but the paper does not elaborate on under which conditions this assumption is reasonable.
   3) Section 4 is extremely difficult to follow because of the large number of definitions. Additional intuition on the roles of the terms in $\mathcal{L}'_d$, $\mathcal{L}_d$, $\mathcal{L}_t$ would greatly improve the readability.

2) **Soundess**
   1) The definition of the optimal representation and the statement in proposition 2 seem not to be applicable to continuous $\bf z$ and $\bf y$ without further clarifications. The value of the (differential) entropies $\mathcal{H}({\bf Y}|{\bf Z})$ and $\mathcal{H}({\bf Z}|{\bf Y})$ approach $-\infty$ whenever there exists an invertible mapping ${\bf y}=f({\bf z})$. As a result, I believe the proof for proposition 2 needs to be revised.
   2) The existence of a homeomorphic mapping between $\bf z$ and $\bf y$ and definition 1 rely on the existence of an underlying mapping ${\bf y} = f({\bf x})$. In other words, this assumption seems to restrict the setting to targets $\bf y$ that can be fully determined from $\bf x$. This aspect is not directly discussed in the main text and the real-world experiments consider distributions in which $p({\bf y}|{\bf x})$ has non-negligible aleatoric uncertainty (super-resolution, depth-estimation, age estimation).


3) **Experiments**
   1) No measure of standard deviation is reported, which makes it difficult to assess the significance of the reported results.
   2) The authors mention that "several widely used regularizers like weight decay and dropout effectively reduce the last hidden layer's intrinsic dimension", but these simple baselines are not included in any comparison.



## Minor Issues
4)  **Presentation**
    1) The distribution $\mathcal{D}$ defined in theorem 2 seems to depend on the specific sample ${\bf y}_i$ but the index $i$ is dropped in the notation.
    2) Figure 2b is helpful in supporting section 4, but I was unable to parse the plot on the right side since there is no direct reference to it in the description.
    3) The differences between plots (b) to (e) in Figure 3 are difficult to relate to the description in the main text since the plots appear quite similar to each other.
    4) Section 5.2 introduces a large number of abbreviations without previous mentions.

### References
[1] Alemi, Alexander A., et al. "Deep variational information bottleneck." arXiv preprint arXiv:1612.00410 (2016).

**Questions:**

1) Can the proposed method be extended to stochastic representations? What is the benefit of considering only deterministic encoders?

2) Under which conditions is the assumption used in Theorem 3 (uniformity) justified?

3) Does the proposed theory address solely tasks with negligible aleatoric uncertainty? In tasks such as super-resolution, it seems that the predictive distribution $p({\bf y}|{\bf x})$ could map a low resolution ${\bf x}$ into multiple possible "correct" outputs ${\bf y}$. Does that mean that an "optimal representation" of $\bf x$ does not exist for this task? How could a homeomorphic mapping between $\bf y$ and $\bf z$ exist in these settings?

4) How are the values for $\lambda_t$ and $\lambda_d$ determined?

5) How consistent are the results reported in Tables 1, 2, 3, and 4 across multiple runs?

---

> ### Author Response · Authors · 2023-11-22
>
> Dear Reviewer awEv,
>
> Thank you for your insightful and detailed comments.
>
> For the concerns:
>
> **Uncertainty, proposition 2 and definition 1 (Soundness & Question 3):** Thanks for these very valuable points for Proposition 2 and Definition 1. You are correct. The defined optimal representation and the corresponding homeomorphism are currently limited to the setting with negligible aleatoric uncertainty. When considering aleatoric uncertainty, i.e., $\mathcal H(Y|X)>0$, the ‘optimal representation’ exists but can not be obtained under our current definitions, as $\mathcal H(Y|Z)$ can never equal 0. By the way, Proposition 2 and Definition 1 hold under epistemic uncertainty.
>
> Proposition 2 and Definition 1 show the homeomorphism between the target and the feature spaces in the limited setting, which currently serves as a hint to explain why we should encourage the topological similarity between the target and the feature spaces. We will extend Proposition 2 and Definition 1, and give some discussion accordingly.
>
> **Extended to stochastic representations (Clarity 1 & Question 1):** Appreciate for pointing this out. Our method can be extended to stochastic representations if $p(z|x)$ follows some fixed distributions like the standard Gaussian distribution which is commonly exploited in VAE. The deterministic mapping assumption in theorem 1 is to cancel the term $\mathcal H(Z|X)$. If $p(z|x)$ follows some fixed distributions, then $\mathcal H(Z|X)$ is a constant and thus also can be canceled.
>
> **Uniform distribution assumption in Theorem 3 (Clarity 2 & Question 2):** We follow [Ghosh & Motani (2023)] to make this assumption. Thanks for this question. We realized this assumption is unnecessary: the entropy of uniform distribution has the largest entropy over all distributions over the support $\mathcal M_i$, thus $\mathcal H(Z|Y)$ is bounded by the right part of Eq 4, which also suggests reducing the intrinsic dimension for a lower $\mathcal  H(Z|Y)$.
>
> **Report standard deviation & include baselines like weight decay& difficult to follow Section 4 (Experiments 1, 2 & Clarity 3):** Thanks for pointing these out. We will revise our paper accordingly.
>
> **How are the values for $\lambda_t$ and $\lambda_d$ determined (Question 4):** Their values are mainly determined by the the value of the task loss. For a high value task loss,  $\lambda_t$ and $\lambda_d$ should also be set to high values.
>
> **How consistent are the results reported in Tables 1, 2, 3, and 4 across multiple runs? (Question 5)**: The standard deviation is reported in Table 1; we will consider reporting standard deviations for experiments on real-world tasks, i.e., Table 2,3,4. In particular, age estimation results are inconsistent, which should be due to the large variance of the ‘few’ subset.

---

### Official Review · Reviewer_pU4H · 2023-10-31

**Soundness:** 2 fair
**Presentation:** 2 fair
**Contribution:** 2 fair
**Rating:** 5
**Confidence:** 3

**Summary:**

This work considers deep learning models trained for regression from the perspective of information bottleneck and the topology of the model’s latent space. The work notes that the concept of the information bottleneck (IB) imposes constraints on the relationship between the latent space of a model and the target space. Through a number of propositions and theorems, the work claims to show that (i) the latent space of a regression model should have the same intrinsic dimension as the target space and (ii) the latent space of a regression model should have the same topology as the target space. To nudge models in this direction, the work develops regularizers that encourage these properties in the latent space. Finally, the work describes both synthetic and real-data experiments that suggest that these regularizers are both helpful for building robust regressors.

**Strengths:**

- **Advancing the science of deep learning for regression:** The fact that so much research into the science of deep learning has focused exclusively on classification at the expense of other tasks (such as regression), is a weakness of the field. This work which investigates IB, intrinsic dimension, and the topology of latent space specifically for models trained to perform regression thus represents a welcome research direction.
- **Utilizing interesting mathematics:** The paper brings together interesting mathematics (topology) with ideas about the information bottleneck. Though some of the formalism of that connection needs to be ironed out, the underlying idea is interesting and seems worth pursuing.

**Weaknesses:**

**Writing correctness and clarity:** There were a number of issues with the writing that made it more challenging to read the work than it should have been. For instance, typos such as:
1. In the introduction, “The homeomorphic between two…” $\mapsto$ “The homeomorphism between two…”.
2. In the introduction, “…and in the topology view,…” $\mapsto$ “…and from the topological viewpoint,…”
3. In Section 5.1, “In contrast, naively lowering the intrinsic dimension ($+L’_{d}$ ) performs poorly and even worse than the baseline, i.e., Tours” $\mapsto$ "...i.e., torus.”

There were also a number of mathematical statements that were made that either didn’t make sense or were not true. For example:
1. The sentence “The continuity represents the $0$th Betti number in topology…” is incorrect. The $0$th Betti number captures the number of connected components in the topological space but has little to do with continuity (otherwise). For instance, a set of three unique (and discrete) points in $\mathbb{R}^3$ has $0$th Betti number $3$ and three disjoint spheres in $\mathbb{R}^3$ also has $0$th Betti number $3$.
2. Pretty much all topological data analysis is based on algebraic topology, so the word ‘algebraic’ in ‘algebraic topological data analysis’ is unnecessary.
3. “However, unlike classification, regression’s target space is naturally a topology space, rich in topology information crucial for the detailed task.” It isn’t clear why the target space of a classification task is less naturally a topological space. Perhaps what the paper means is that there is less topological structure in the discrete target space of classification tasks. Then again, if one is only using the $0$th homology groups (as this paper does), one is not capturing any higher dimensional structure anyway.
4. Proposition 2 concerns continuous maps between $\mathbf{Z}$ and $\mathbf{Y}$. What are $\mathbf{Z}$ and $\mathbf{Y}$? Continuous maps only make sense when one has a defined topology on the domain and target space, this doesn’t seem to currently exist in the work.

There were also a few cases where notation was used that was never defined.
1. What is $\mathcal{Y}$ in Theorem 2?
2. If $\mathbf{X}$, $\mathbf{Z}$, and $\mathbf{Y}$ are going to be used in proofs, it should be stated what they are.

Finally, the reviewer had a hard time understanding some of the figures
1. Figure 1(b) is used to illustrate a point about intrinsic dimension in the introduction. The reviewer had trouble understanding what the task is here and what the reader is meant to see in this scatter plot. More context should probably be added.

**Use of the word ‘topological’ when only $H_0$ is used:** In the cases considered in this paper, the only topological property that $H_0$ measures is the number of connected components of a space (it has been shown that persistent homology also captures geometric information like curvature, so such features may also be leaking into regularization). As such, it captures only a small fraction of the total topological characterization of the space (particularly for high-dimensional spaces). It is this reviewer’s opinion that this fact should be more explicitly stated than what is currently found on the bottom of page 5. For instance, it would be more accurate to call this ‘connectedness regularization’. It is probably not necessary to actually make this shift, but it does highlight that what is being advertised in much of the work is only marginally captured by the actual algorithms. Also, the paper makes a point of noting that regression tasks have richer topology than classification tasks, but if one is only using $H_0$, the target spaces are effectively the same.

**Concern about where improvements are coming from:** One interpretation of the proposed method is that by constraining topology and intrinsic dimension (even on $\mathbf{Z}$) based on ground truth, one is imposing built-in priors to the regression model. With this extra information, it is not surprising that the new model achieves better performance. One way to test this would be to impose the same regularization on model outputs in $\mathbf{Y}$ rather than $\mathbf{Z}$. Are the same improvements seen? I am not sure that the results are less interesting, but the connection to IB might be less compelling.

**Questions:**

- How does this regularization method scale to higher-dimensional target spaces?
- Is it possible to incorporate higher homology groups into the regularization terms?
- Do intrinsic dimension estimators outside of the approach leveraged in the regularizer capture a decrease in intrinsic dimension of the latent space representations?
- How important is it to apply these regularizers to the latent space? Would a similar effect be seen if the outputs were regularized instead?

---

> ### Author Response · Authors · 2023-11-22
>
> Dear Reviewer pU4H,
>
> Thank you for your insightful and detailed comments.
>
> For the weaknesses:
>
> **Writing correctness and clarity**:  Thanks for the detailed remarks. We will revise our paper accordingly. We should define the topology for the target and latent space as the topology induced by the metric, Euclidean distance. Since regression’s target space is naturally a metric space (while classification’s is not), the topology of the target space for classification is harder to exploit than regression.
>
> **Use of the word ‘topological’ when only $H_0$ is used**: While our designed regularizer only consider $H_0$, our theoretical analysis is not limited to $H_0$. Besides, it of course can consider higher homology groups, and some methods already exist, like the RTD [arxiv 2302.00136]. The only reason why we just consider $H_0$ is that we exploit the topological autoencoder as our topology part $L_t$, and higher (1-dimensional) topological features merely increase runtime for topological autoencoder.
>
> **Concern about where improvements are coming from**: Thanks for this interesting point. 1) The topology and intrinsic dimension can be regarded as built-in priors, and we are imposing the two built-in priors on the regression model. 2) The model outputs, i.e. predicted Y’, can also be regarded as the representation Z, where the intrinsic dimension is commonly forced to be the same as Y, and the MSE loss or other loss clearly encourages Y’ and Y to be the same (including the topology aspect).
>
> For the questions:
>
> **Scale to higher dimensional**: Our designed regularizer is based on the topological autoencoder and Birdal’s regularizer. Both methods work well for high-dimensional space, e.g., dimension equals the number of pixels for image input. Based on their good performance when scaled to high dimensional, our method should work well for high-dimensional target space.
>
> **incorporate higher homology groups**: of course can. Please see above the weaknesses part.
>
> **decrease in intrinsic dimension of the latent space representations**: Do you mean whether the intrinsic dimension of the latent space really decreased? Based on the visualization and the estimated intrinsic dimension estimated by Birdal’s estimator, the intrinsic dimension is decreased. For exploiting other intrinsic dimension estimators, we will report these results in our future submission.
>
> **apply to the outputs**:  please see above the weaknesses part.

---

### Official Review · Reviewer_5Zit · 2023-11-01

**Soundness:** 2 fair
**Presentation:** 2 fair
**Contribution:** 2 fair
**Rating:** 3
**Confidence:** 3

**Summary:**

The paper studies the IB principle for regression problems. Then, the connection between the IB principle and topological properties is established. Basing on these observations, the PH-reg regularizer is proposed to harmonize topology of Z space and target space Y and to minimize the intrinsic dimension of latent space Z. Finally, experiments with both synthetic and real-world datasets show that the proposed PH-reg brings some improvement in quality measures.

**Strengths:**

1. Only a few papers study application of topology to machine learning and only a few papers study differentiable topological losses.
2. The paper is mostly well written and clear.
3. New theoretical results are presented.
4. Experiments show improvements in quality measures for super-resolution, depth estimation and age prediction problems.
Ablation studies are provided.

**Weaknesses:**

1. I have concerns about the IB principle. Neural networks are deterministic functions, and, thus H(Y|Z) = 0 always.
H(Z|Y) > 0 when different Z map to the same Y. This can happen when different X map the the same Y, because for real large networks different input objects X have different embeddings Z. So, some H(Z|Y)>0 depends only on the dataset itself, not the network for realistic scenarios.

2. $min_Z \left( I(Z,X) - \beta I(Z,Y) \right) $ and $min_Z \frac{I(Z,X)}{ \beta I(Z,Y) }$ are different problems. Thus, Theorem 1 is wrong.

3. The regularizer in eq. 8 is the topological loss from Moot et. al, 2020. You should explicitly mention it.

4. Sometimes the narration is clumsy, ex.: "Figure 1(a) provides a t-SNE visualization
of the 100-dimensional feature space with a ’Mammoth’ shape target space."
Here I don't understand what does 100-d space mean and what is "Mammoth target space".
Also, I think for your case visualization with t-SNE is not a good choice since t-SNE often tears a manifold apart.
Try to use PCA.

minor:
1. Caption to Figure 3: model' - typo
2. In Section 3, you write "Consider a dataset S = {xi, yi} with N samples xi , which typically is an image (xi \in Rdx1×dx2 )"
color channels of image are missing.

**Questions:**

1. what do functions f1, f2, f3, f4 from Section 5.1 mean?
2. I don't understand the purpose of experiments from Section 5.1.
You embed 3D point cloud into a high dimensional space by adding 96 extra noise coordinates.
Then, you try to predict 3 original coordinates, right?
So, ideally, the encoder must learn to ignore 96 noise coordinates and learn identity mapping for the rest of them.
When you enforce the topological regularizer from (Moor et. al, 2020), you enforce pairwise distances in input and target spaces to be the same. Obviously, noise coordinates are forced to be ignored.
3. What is the target space in the age prediction problem, R ? Then, the homeomorphism from image space R is not possible.
4. What are the target spaces and Z space for depth estimation and super-resolution problems?
5. For experiments with DIV2K, the improvement is very small (~0.1%-0.2%). Also std. intervals are not provided.
Have you optimized hyperparams on the validation dataset and evaluated final quality on the test set?
Otherwise, results are not valid.

---

> ### Author Response · Authors · 2023-11-22
>
> Dear Reviewer 5Zit,
>
> Thank you for your insightful and detailed comments.
>
> For the weaknesses:
>
> **W1. $\mathcal H(Y|Z) = 0$ always. $\mathcal H(Z|Y) > 0$ depends only on the dataset itself:**
> 1) There is a misunderstanding between the predicted Y’ by neural networks and the target Y. For predicted Y’, $\mathcal H(Y’|Z) = 0$ always, since neural networks are deterministic functions. But this does not hold for $\mathcal H(Y|Z)$.
> 2) It seems you mean different X -> different Z -> the samy Y, thus $\mathcal H(Z|Y) > 0$ and this depends only on the dataset itself. This should be correct for general purpose representation learning, which targets a wide range of downstream tasks. However, for a specific task, in an extreme case, you can treat the predicted Y’ as the representation Z, which clearly shows $\mathcal H(Y’|Y) =0$ is desirable and is the learning target. By the way, from the invariance representation learning point of view, lowering $\mathcal H(Z|Y)$ is learning invariance representation with respect to Y.
>
> **W2. minimizing the difference is not equivalent to minimizing the ratio:**  Thanks for pointing this out. We will revise Theorem 1.
>
> **W3. Should explicitly mention the topology autoencoder:** Thanks for pointing this out. We will emphasize it in the main paper.
>
> **W4. Try PCA:**  Appreciate for pointing this out. The PCA works better than t-SNE for visualization on the synthetic dataset. The ‘100-d space’: the feature space is 100 dimensional. The ‘Mammoth target space’: the target space is a mammoth. Some descriptions are given in Section 5.1.
>
> For the questions:
>
> **Q1. Meaning of functions f1-4:**  Functions f1-4 are functions to encode the coordinates. Detailed expressions for these functions can be found in Appendix B.
>
>
> **Q2. experiments from Section 5.1:** It is not learning identity mapping, as the original coordinates are encoded by f1-4. The key purposes are: 1) show the feature space is topologically similar to the target space and regression potentially captures the topology of the target space. 2) verify our methods. The objects in Section 5.1 are topologically different, and experiments with those objects can clearly verify our methods. For real-world tasks, we lack datasets with different ‘shape’ target space.
>
> **Q3. target space of age prediction:** The target space for age estimation is discrete points. Besides, the homeomorphism is between Z and Y rather than between image space and Y.
>
> **Q4. target spaces of depth estimation and super-resolution:** Depth estimation: 1-dimensional line; super-resolution: 3-dimensional space.
>
> **Q5. Experiments on DIV2K**
> :  1) Most methods have ‘similar’ performance with DIV2K, and the improvements for many other methods targeted at super-resolution are also very small. The small improvement should be due to the task/dataset itself. 2) Usually, the computational cost is high for super-resolution, but we will consider reporting the std and intervals if possible. 3) We follow the setting of EDSR for the hyper-parameters, apart from the $\lambda_t$ and $\lambda_d$.

---

### Official Review · Reviewer_5oRh · 2023-11-02

**Soundness:** 2 fair
**Presentation:** 3 good
**Contribution:** 2 fair
**Rating:** 3
**Confidence:** 4

**Summary:**

This paper considers implementing information bottleneck (IB) principle for regression problem. To this end, authors first derive a new generalization error bound, showing that in regression scenario, the generalization error is upper bounded by a function of the conditional entropy of $Z$ given $Y$, i.e., $H(Z|Y)$, in which $Z$ is the learned feature. Given this, authors mention the connection between $H(Z|Y)$ with respect to the intrinsic dimension of $Z$. Hence, to control the generalization error, it makes sense to minimize the intrinsic dimension of $Z$.

Further, based on the optimization representation condition that $H(Y|Z)=H(Z|Y)$, authors emphasized the necessarity to add an additional topology preserving regularization terms. Experiments are conducted on three real world data to demonstrate the effectiveness of the two regularization terms.

**Strengths:**

1. The IB principle for regression problem is less considered. The motivation of this work is promising.
2. The new generalization error bound in Eq.~(2) is interesting, although it still has some issues (see below).

**Weaknesses:**

Overall, I have some concerns regarding Theorems 1, 2 and 3, especially Theorem 2.

1. I have several concerns regarding the new generalization error bound in Eq.~(2).

1.1 how this bound is connected to the new bound in [1], which emphasized the role of I(X;Z|Y) in classification tasks.

[1] Kawaguchi, Kenji, et al. "How Does Information Bottleneck Help Deep Learning?." arXiv preprint arXiv:2305.18887 (2023).

1.2 Is the bound tighter or not? or does H(Z|Y) a good indicator on the generalization error in regression problems? Can you validate this point?

1.3 In fact, the Theorem 2 is a bit unclear, especially the description on Q. Are there some properties that Q should have?

1.4 It is unclear how to derive from Eq.(28) to Eq.(29). I understand that $(E(|z-\bar{z}|_2))^2 \leq E |z-\bar{z}|_2^2$. However, I do not understand why $E |z-\bar{z}|_2^2$ can be further bounded by $Q(H(Z|Y))$. Does $Q$ exist for general case? In fact, authors only illustrated that $Q$ exists for Gaussian case and uniform case. I am not sure if the derivation from Eq.(28) to Eq.(29) holds for general case.

2. Regarding Theorem 1, it only holds for a deterministic mapping function from $X$ to $Z$, since you assume that $H(Z|X)=0$? This should be emphasized in the main paper, since majority of deep IB approaches use stochastic representations and $H(Z|X)\neq 0$.
Actually, this also decreases the applicability of Theorem 1.

3. Theorem 3 is actually a straightforward result of [Ghosh & Motani (2023), Proposition 1]. This should also be emphasized in the main paper.

4. Apart from the above concerns, I also feel the two regularization terms are not novel.
In fact, the connection between entropy and intrinsic dimension has been carefully discussed in [Ghosh & Motani (2023)].
The way that authors evaluate the intrinsic dimenion seems to be directly from Birdal et al. (2021).
As for another regularization term on topology perserving, it also shares rather similar to that in (Moor et al., 2020).

5. Finally, regarding the experiments. It would be much helpful if authors can compare their method to other prevalent deep IB approaches. For example, nonlinear information bottleneck also considers a regression setup. Another example is the convex information bottleneck [2].

[2] Rodríguez Gálvez, Borja, Ragnar Thobaben, and Mikael Skoglund. "The convex information bottleneck lagrangian." Entropy 22.

**Questions:**

please see weaknesses 1, 4, and 5.

---

> ### Author Response · Authors · 2023-11-22
>
> Dear Reviewer 5oRh,
>
> Thank you for your insightful and detailed comments.
>
> For the weaknesses:
>
> **W1.1 & 1.2 & 1.3. Connections with the bound in [1], and illustrations of the function Q**: The bound in [1] emphasizes two terms, i.e., $I(X; Z|Y)$ and $I(\phi; S)$, where the second term measures the effect of overfitting the encoder $\phi$. $S$ here represents the dataset. By contrast, our bound only emphasizes one term, i.e., $\mathcal H(Z|Y)$, which equals $I(X; Z|Y)$ with deterministic encoders:
> > $I(X; Z| Y) = I (X; Z) - I(Y; Z) $(shown in [1]) $= H(Z) - H(Z|X) - (H(Z) - H(Z|Y)) =  H(Z|Y) - H(Z| X) =  H(Z|Y)$.
>
> Note, the linear dependence on H(Z| Y, X) in the bound [1] is also mainly for deterministic encoders. For stochastic representations, $H( Z| X, Y) \approx 0$ if the injected noise is small, and thus the linear dependence also holds.
>
> The tightness of our bound is determined by the function Q, which aims bound the dispersion/standard deviation of a distribution by its entropy. Compared with the bound in [1], with the Q suggested in proposition 1,  bound [1] is tighter than ours if we ignore the term $I(\phi; S)$. Because the Q we suggested is exponential dependence while their bound is linear dependence. Q exists for general cases, as the dispersion/standard deviation and the entropy commonly can be estimated for a specific distribution. We thus can find a function Q to scale the entropy to larger than its dispersion/standard deviation.
>
> It is worth mentioning that we are not targeting a tight/advanced bound. The bound is mainly introduced to support our claim: ‘minimizing $\mathcal H(Z|Y) $can be seen as learning a minimal representation by reducing noise. The minimality can reduce the complexity of Z and prevent neural networks from overfitting.’ Comparing it with other advanced bounds and demonstrating it is a good indicator are somehow out of our scope. We will take these points into consideration to revise the manuscript.
>
>
> **W1.4. From Eq.(28) to Eq. (29):**
> Because $z$ in Eq. (28) follows the distribution $(Z| Y=y_i)$, and based on Eq. (14) we can thus bound $E ||z-\bar z||_2$ by $\mathcal H(Z|Y=y_i)$.
>
> **W2. deterministic mapping between Z and X**: Appreciate for pointing this out. We will emphasize this in the main paper. Although our analysis is based on IB, we target neural networks, which are commonly deterministic functions. Besides, our method can be extended to stochastic representations if $p(z|x)$ follows some fixed distributions like the standard Gaussian distribution, which is commonly exploited in VAE. The deterministic mapping assumption in theorem 1 is to cancel the term $\mathcal H(Z|X)$. If $p(z|x)$ follows some fixed distributions, then $\mathcal H(Z|X)$ is a constant and thus also can be canceled.
>
> **W3.** Thanks for pointing this out. We will emphasize this in the main paper
>
> **W4. The novelty of the two regularizer terms**: Our main contribution is the two established connections between the topology and IB for regression representation learning, and the two regularizer terms are mainly introduced to verify the connections. Besides, although the two regularizers are designed based on Birdal’s regularizer and the topological autoencoder, designing the regularizer in such a way, that follows our established connections, is novel.
>
> **W5. Compared to other IB approaches.** Thanks for the suggestion. We will revise the manuscript accordingly.

---

### Official Review · Reviewer_7gBF · 2023-11-09

**Soundness:** 1 poor
**Presentation:** 2 fair
**Contribution:** 2 fair
**Rating:** 3
**Confidence:** 4

**Summary:**

The paper proposes to use a regularizer PH-Reg, with aim to lower the intrinsic dimension of the feature space while trying to preserve the "0-dimensional topologically relevant distances", in the context of representation learning. Some arguments, favoring  reducing a quantity which could be related to the intrinsic dimension of the feature space, in terms of information bottleneck approach, are given. Some  empirical verifications of the method are described, an access to the source code for experiments for reproducibility purpose is not provided.

**Strengths:**

The paper tries to relate quantitative characteristics of data representations from information theory with topological data characteristics, following several recent approaches.

**Weaknesses:**

1) The proposed "intrinsic dimension lowering" loss term $\mathcal{L}_d$  actually disturbs the intrinsic dimension in an unclear way, it can both increase and decrease it, since, for a given data representation $Z$, the term   $\mathcal{L}_d$  involves the ratio of logarithms $\log E(Z_n)/ \log E(Y_n)$ where $Y$ is another data representation. There is no $\log E_n(Y)$, $Y-$dependent part, in the formula for intrinsic dimension of $Z$, see eg arXiv:2306.04723 or J. M. Steele, Growth rates of euclidean minimal spanning trees with power weighted edges, The Annals of
Probability, 16(4):1767–1787, 1988.  So the paper narrative concerning the lowering of intrinsic dimension is seemingly in contradiction with the reported experiments setup.

2) Reproducibility check concerning the reported experiments in the manuscript (Section 5) could not be performed. No source code was made available during the review phase.

3) Theoretical results reported in Section 3 seem to have strong intersection with previously  published papers, in particular, Theorem 3 from Section 3 seems to be similar to Proposition 1 from  [Ghosh & Motani (2023)], cf Appendix A.3.

4) The definition of the intrinsic dimension employed in Section 3, which is the same as the definition in  [Ghosh & Motani (2023)],  is completely different from the definition of intrinsic dimension via the 0-th persistent homology, employed in Section 4. So, a priori, there might be no connection between the two quantities.

3) Training details are  lacking in the description of the experiments reported in Table 1, Table 2, Table 3, Table 4.   In particular, how many iterations/epochs were used, what were the stopping criteria? It might be that the experiments, if they can be reproduced, cf above, could be explained by overfitting of the baselines, since it seems that the standard regularizers, such as weight decay, were not used.

4) Standard deviations are not reported in Table 2, Table 3, Table 4.

5) Only a single baseline method is employed for comparison in each case in Table 2, Table 3, Table 4.

6) The loss $\mathcal{L}_t$ employed "to preserve topology", has been recently shown to have the following drawbacks: firstly, this loss is not continuous, secondly, moreover, diminishing   $\mathcal{L}_t$  can lead to bigger difference in topology, cf  arxiv2302.00136, Appendix J, Figure 10.

7)  Related work section does not mention many relevant papers, eg arXiv:2106.04024, arXiv:2201.00058, arXiv:2306.04723, J. M. Steele, Growth rates of euclidean minimal spanning trees with power weighted edges, The Annals of
Probability, 16(4):1767–1787, 1988.

9) The writing could be improved, there are some vague statements and not very clearly defined notions.

Below are some specific remarks:

page 2 : "intrinsic dimension of the feature space" - what is this ? From what follows a reader can guess that seemingly it is  what can be  called intrinsic dimension of the dataset points  in the feature space, is it?

page 2 :  Figure 1a - The right picture with supposedly "topology similar" has clearly too many clusters, eg a separate leg cluster, and, on the contrary, it does not preserve topology. The preserving topology representations exist in the literature for this mammooth dataset and they look completely different.

page 2 : Figure 1b "Lower the intrinsic dimension" picture - Is the paper saying that a good representation of the 3d mammoth for regression task should be 1dimensional? First of all, the 3d object smashed into 1d space does not seem to be a good representation for the vast majority of tasks including many other regression tasks.  Secondly, such a smashing into 1d space representation would depend heavily on the regression task, which is also usually considered to be bad for a data representation.

page 2:  "The homeomorphic between two spaces" - should it be homeomorphism ?

page 2:  " t-SNE visualization of the 100-dimensional feature space" - the 100d feature space is described only in Section 5.1, several pages down. It would be better to mention it here on page 3, otherwise it is impossible to understand what are the 100 dimensions mentioned here.

page 2: "we are the first to explore topology in the context of regression representation learning"- actually, the autoencoders learn the regression task of reproducing the coordinates of the data points, so no, the topology has been heavily explored in previous works in the context of regression representation learning.

page 3: "The intrinsic dimension of the last hidden layer" -  what is this ? From other contexts a reader can probably guess that, seemingly, it is  what can be called intrinsic dimension of the dataset points  in the last hidden layer, is it?

page 3:  "1-dimensional (Trofimov et al 2023) topologically relevant distances" - actually the method from (Trofimov et al 2023) preserves both 0-dimensional and 1-dimensional topological features, and even arbitrary $k-$dimensional features.

page 3:  "However, unlike classification," - as already mentioned, the autoencoders learn the regression task of reproducing the coordinates of dataset points.

page 4: "$Dim_{ID} M_i$ is the intrinsic dimension of the manifold"(corresponding to the distribution)   - which definition of intrinsic dimension is used here?

page 5: "is a homomorphism between ... and" - is a homomorphism from... to ...

page 5: "that the set of points S sampled from" - that the set of points S _is_ sampled from ?

**Questions:**

What is the relation between the quantity called "intrinsic dimension" in the Theorem 3, Section 2, seemingly based on ϵ-neighborhood intrinsic dimension and the "intrinsic dimension" calculated via asymptotic of the minimal spanning trees in Section 3?

---

> ### Author Response · Authors · 2023-11-22
>
> Dear Reviewer 7gBF,
>
> Thank you for your insightful and detailed comments.
>
> For the weaknesses:
>
> **W1. The term $\mathcal L_d$ can both increase and decrease the intrinsic dimension:** Yes, this term can do both.  We aim to encourage a feature space with an intrinsic dimension (ID) equal to the target space rather than only lowering the ID. Therefore, we apply $\mathcal L_d$ to increase or decrease as necessary.  We will revise the manuscript to state this more explicitly.
>
> **W2. Release the Code**: We will release the code upon paper acceptance.
>
> **W3. Theorem 3 Intersection with [Ghosh & Motani (2023)]**: Theorem 3 is a result of [Ghosh & Motani (2023), Proposition 1]. This is shown in the appendix A.3, and we will emphasize it in the main paper.
>
> **W4. Different definitions for intrinsic dimension**:  Yes, the definitions for the 0-th persistent homology (employed in the regularizer) and the $\epsilon$-neighborhood intrinsic dimension (employed in our theoretical analysis)  are different. However, this difference does not invalidate our conclusions, as our theoretical analysis is purely based on the $\epsilon$-neighborhood intrinsic dimension. In addition, although the 0-th persistent homology has a different mathematical definition and its value can be different, it is also defined for the intrinsic dimension. It is thus reasonable to exploit the 0-th persistent homology to constrain the intrinsic dimension.
>
> A similar concern with corresponding discussions can be found here: https://openreview.net/forum?id=099uYP0EKsJ&noteId=zMiYTAyu9ol .
>
>
> **W5, 6, 7, 9, 10. Lacking training details, standard deviations, and related works; only a single baseline;  writing could be improved**: Thanks for the detailed remarks. We will revise the manuscript accordingly.
>
> **W8. Drawbacks of the topology autoencoder**
> **A**: We also noticed this and once considered the RTD [arxiv 2302.00136]. However, the topology autoencoder works well in reality, and its drawbacks do not invalidate our conclusion.
>
> **We thank you for your detailed remarks. We want to clarify the remark 2 (about the Figure 1a) and the remark 3 (about the Figure 1a):**
> - **Figure 1a, topology similar**: We tried the PCA as suggested by reviewer 5Zit, and found the ‘too many clusters’ is due to the t-SNE, and the results obtained by PAC do not present such ‘clusters’.
> - **Figure 1b, 1d space for 3d mammoth**: This figure is not for 3d mammoth. It is for the depth estimation task, where the target space is 1d.
>
> **Question:** Please see the response for the W4.

---

### Author Response · Authors · 2023-11-22

Dear Reviewers, dear AC,

We do thank you and it is a pleasure to have your high-quality comments. We decided to withdraw the paper for future submission. We hope we have addressed your concerns, and please let us know if you have anything you want to discuss.

Thank you again to all the reviewers and AC.

&nbsp;

Yours sincerely,

Authors